# Dissemination of pathogenic bacteria is reinforced by a MARTX toxin effector duet

Sanghyeon Choi [1,2,11], Youngjin Lee[2,11], Shinhye Park[2,3], Song Yee Jang[2,4], Jongbin Park[2], Do Won Oh[2,5], Su-Man Kim[2,6], Tae-Hwan Kim[2,7], Ga Seul Lee[4,8], Changyi Cho[9], Byoung Sik Kim [9], Donghan Lee[10], Eun-Hee Kim[10], Hae-Kap Cheong[10], Jeong Hee Moon [4], Ji-Joon Song [1], Jungwon Hwang [2] ✉ & Myung Hee Kim [2] ✉

Multiple bacterial genera take advantage of the multifunctional autoprocessing repeats-in-toxin (MARTX) toxin to invade host cells. Secretion of the MARTX toxin by *Vibrio vulnificus*, a deadly opportunistic pathogen that causes primary septicemia, the precursor of sepsis, is a major driver of infection; however, the molecular mechanism via which the toxin contributes to septicemia remains unclear. Here, we report the crystal and cryo-electron microscopy (EM) structures of a toxin effector duet comprising the domain of unknown function in the first position (DUF1)/Rho inactivation domain (RID) complexed with human targets. These structures reveal how the duet is used by bacteria as a potent weapon. The data show that DUF1 acts as a RID-dependent transforming NADase domain (RDTND) that disrupts $NAD^+$ homeostasis by hijacking calmodulin. The cryo-EM structure of the RDTND-RID duet complexed with calmodulin and Rac1, together with immunological analyses in vitro and in mice, provide mechanistic insight into how *V. vulnificus* uses the duet to suppress ROS generation by depleting $NAD(P)^+$ and modifying Rac1 in a mutually-reinforcing manner that ultimately paralyzes first line immune responses, promotes dissemination of invaders, and induces sepsis. These data may allow development of tools or strategies to combat MARTX toxin-related human diseases.

Multifunctional autoprocessing repeats-in-toxin (MARTX) toxins, secreted by multiple Gram-negative bacteria, are primary virulence factors that promote initial colonization, dissemination, and lethality in a wide range of hosts, including humans[1–4]. Among bacterial genera, *Vibrio* is the main genus of bacteria that secrete MARTX toxins[1,5]. These single-polypeptide exotoxins are equipped with a diverse repertoire of enzymatically functional effector domains arranged in a modularly structured fashion within their central regions[6]. Once MARTX toxins translocate into host cells, the toxins are cleaved to release functionally independent effector domains that target specific host substrates[1,2,7,8].

[1]Department of Biological Sciences, Korea Advanced Institute of Science and Technology (KAIST), Daejeon 34141, Korea. [2]Microbiome Convergence Research Center, Korea Research Institute of Bioscience and Biotechnology (KRIBB), Daejeon 34141, Korea. [3]Department of Microbiology and Molecular Biology, Chungnam National University, Daejeon 34134, Korea. [4]Core Research Facility & Analysis Center, KRIBB, Daejeon 34141, Korea. [5]Graduate School of Medical Science and Engineering, KAIST, Daejeon 34141, Korea. [6]Department of Biology Education, Chonnam National University, Gwangju 61186, Korea. [7]College of Veterinary Medicine, Chungnam National University, Daejeon 34134, Korea. [8]College of Pharmacy, Chungbuk National University, Cheongju, Chungbuk 28644, Korea. [9]Department of Food Science and Biotechnology, Ewha Womans University, Seoul 03760, Korea. [10]Korea Basic Science Institute, Cheongju, Chungbuk 28119, Korea. [11]These authors contributed equally: Sanghyeon Choi, Youngjin Lee. ✉e-mail: jwhwang@kribb.re.kr; mhk8n@kribb.re.kr

*Vibrio vulnificus* is the most deadly of the foodborne pathogens, with a case fatality rate of 50%[9]. *Vibrio* spp. survive within a hostile environment and expand their virulence by exchanging or rearranging the genetic information encoding the effectors within MARTX toxins[2,10,11]. By doing this, the clinical strain *V. vulnificus* MO6-24/O generates a MARTX toxin (MARTX$_{Vv}$) containing four effectors, which are arranged in the following order: domain of unknown function in the first position (DUF1), the Rho inactivation domain (RID), the α/β hydrolase domain (ABH), and a Makes caterpillars floppy-like effector (MCF) with a cysteine protease domain (CPD). The latter is activated by binding to host inositol hexakisphosphate, resulting in cleavage of the linkers that connect the effectors[12]. The functions of most of the effectors within the MARTX toxins of all biotypes of *V. vulnificus* are known; the exception is DUF1. Previously, we reported that DUF1 and RID within MARTX$_{Vv}$ are not uncoupled by activated CPD in host cells; rather, they are discharged as a duet (it is assumed that the DUF1-RID effector module acts as a functional unit[1]).

Successful host invasion by pathogenic bacteria such as *V. vulnificus* relies on the ability of their MARTX toxins to counteract host defense mechanisms, although it is not clear how this is achieved[3,12,13]. The effector RID, which is shared by the MARTX toxins of human pathogens, is an N$^\varepsilon$-fatty acyl-transferase that modifies the lysine residues within the C-terminal polybasic region (PBR) of the small GTPase Rac1 at the cell membrane, thereby inhibiting cellular processes such as phagocytosis and production of reactive oxygen species (ROS)[14]. A previous study shows that RID appears to block Rac1-dependent mitogen-activated protein kinase (MAPK) signaling to suppress pro-inflammatory responses prior to induction of actin cytoskeleton collapse by a *V. cholerae* MARTX toxin (MARTX$_{Vc}$) effector called the actin cross-linking domain (ACD)[13].

ROS production is important in that it activates innate immune responses early in the course of infection. An isoform of nicotinamide adenine dinucleotide phosphate (NADPH) oxidase, called NOX, is the multicomponent enzyme responsible for ROS generation mainly in macrophages and neutrophils[15]. The GTP-bound (active) form of Rac1 or Rac2 acts as a critical regulator of NOX2 activation[15] by interacting with p67$^{phox}$ through its Switch I region[16–18]. The participation of small GTPases in regulating ROS generation plays an important role in innate immune responses against invading microbes[19–22]. It is assumed that MARTX toxin-secreting pathogens utilize the effector RID to disrupt first line immune responses triggered by the Rac GTPase-NOX2 signaling axis; however, we do not know how RID affects this signaling axis, nor do we know anything about the interrelated roles of DUF1 and RID during/after cell invasion.

In this study, we performed interactome analyses and found that the DUF1-RID effector duet hijacks calmodulin (CaM) and Rac1 in infected host cells. Crystal and cryo-EM studies of the effector duet complexed with CaM and Rac1, together with biochemical and immunological analyses, provide key insight into a concerted invasion strategy in which pathogenic bacteria coordinate the DUF1-RID duet, the components of which act synergistically to suppress ROS generation and promote dissemination of pathogens, ultimately resulting in sepsis.

## Results

### The DUF1-RID effector duet hijacks calcium-free CaM
A previous study used mass spectrometry analysis to identify the Rho family small GTPase Rac1 as a target for RID[14]. In separate experiments, we performed an interactome analysis using ectopically expressed inactive RID$_{C/A}$ (residue C2838 mutated to alanine, corresponding to the active residue in the previous study[14]) in HEK293T cells. Note that we constructed RID based on the inositol hexakisphosphate-activated CPD-mediated cleavage sites between effectors within MARTX$_{Vc}$[23]. Unlike in the previous study[14], we found that CaM showed the highest intensity ratio (relative to the control) among the potential RID

interactors, although Rac1 was also detected as an interactor (Supplementary Fig. 1a).

We previously reported that MARTX$_{Vv}$ discharges the effectors DUF1 and RID as a duet (Fig. 1a)[1]. Further analysis of DUF1-RID duet- or RID-carrying MARTX toxins secreted from different bacterial genera revealed a tendency towards the composition of the effectors with genus type (Fig. 1b). Among the different genera, *Vibrio* spp. is the primary genus that secretes MARTX toxins. Bacteria belonging to this genus secreted a similar proportion of DUF1-RID duet- or RID only-containing toxins (Fig. 1b). *V. cholerae* and *V. vulnificus* are the main species within this genus, and the former tended to largely secrete RID-containing toxins while the latter mainly secreted DUF1-RID duet-containing toxins (Fig. 1b). Additionally, opportunistic human pathogens belonging to *Proteus* spp. secreted MARTX toxins carrying RID or its N-terminal domain-deleted form (Fig. 1b). Importantly, the MARTX toxins of most clinical *V. vulnificus* strains carried the DUF1-RID duet, while clinical *V. cholerae* strains contained mainly RID (Fig. 1c). Consistent with CPD-mediated cleavage site-based analysis[23], we found that nearly all RIDs from *Vibrio* spp. comprise an N-terminal domain (ND, RID$_{ND}$), a membrane localization domain (MLD)-containing domain (MLD_C, RID$_{MLD\_C}$), and a catalytic domain (CD, RID$_{CD}$), with few exceptions (Fig. 1a, b). Thus, our previous results, coupled with those obtained herein, led us to question whether CaM is a host target of RID alone, or also a target of DUF1.

First, we evaluated the cytopathogenicity of MARTX$_{Vv}$ DUF1 coupled to inactive RID$_{C/A}$ (i.e., DUF1-RID$_{C/A}$). The results showed that DUF1-RID$_{C/A}$, but not DUF1 alone or RID$_{C/A}$, caused marked morphological changes in HEK293T cells, suggesting that DUF1 induces cytotoxicity in a RID-dependent manner that is unrelated to the Rac1 modification function of RID (Fig. 1d).

Subsequently, the interaction between DUF1-RID and CaM was validated by performing in vitro pull-down assays using purified proteins. The duet interacted with CaM, regardless of the presence of Ca$^{2+}$; indeed, it interacted more readily with Ca$^{2+}$-free CaM (Fig. 1e and Supplementary Fig. 1b). Consistent with the pull-down results, isothermal titration calorimetry (ITC) analysis revealed that the effector duet binds much more strongly to Ca$^{2+}$-free CaM (dissociation constant ($K_D$) = 80 nM) than to Ca$^{2+}$-bound CaM (CaM/Ca$^{2+}$, $K_D$ = 1.93 μM), with a 1:1 binding stoichiometry (Supplementary Fig. 1c–e). It should be mentioned that the binding affinity of MARTX$_{Vc}$ RID ($K_D$ = 4 μM) and MARTX$_{Vv}$ RID ($K_D$ = 20 μM) for Rac1 was much weaker[14], suggesting that CaM is the primary host target of the effector duet. Next, we showed that the CaM-binding domain of RID$_{ND}$ is the most critical part that interacts with CaM (Fig. 1e). The interaction between DUF1-RID$_{ND}$ and CaM was comparable with that of DUF1-RID, while in the absence of RID$_{ND}$, DUF1 failed to interact with CaM. Thus, we named RID$_{ND}$ the "CaM-binding domain (RID$_{CBD}$)". Collectively, these data suggest that the DUF1-RID duet hijacks host CaM via RID$_{CBD}$ to facilitate infection.

### DUF1 is transformed into a NAD(P)$^+$ hydrolase upon RID-dependent CaM-binding
To elucidate the relevance of CaM binding by the effector duet, we first tried to solve the structure of the MARTX$_{Vv}$ DUF1-RID duet in the presence/absence of CaM; however, we were unable to initially generate crystals. Later, we did manage to generate crystals of DUF1-RID$_{CBD}$ (residues, 1959–2374) in the absence of CaM, and determined the structure at a resolution of 3.38 Å (Supplementary Figs. 2a and 3a, and Supplementary Table 1). Subsequently, we determined the structure of DUF1-RID$_{CBD}$ in complex with Ca$^{2+}$-bound CaM or Ca$^{2+}$-free CaM at a resolution of 2.82 Å and 2.90 Å, respectively (Supplementary Fig. 2b, c and 3b–d, and Supplementary Table 1). The structures of both complexes are identical (RMSD of 1.323 Å) (Supplementary Fig. 2d). Thus, from now on we describe the structural details of DUF1-RID$_{CBD}$ complexed with Ca$^{2+}$-bound CaM, unless stated otherwise.

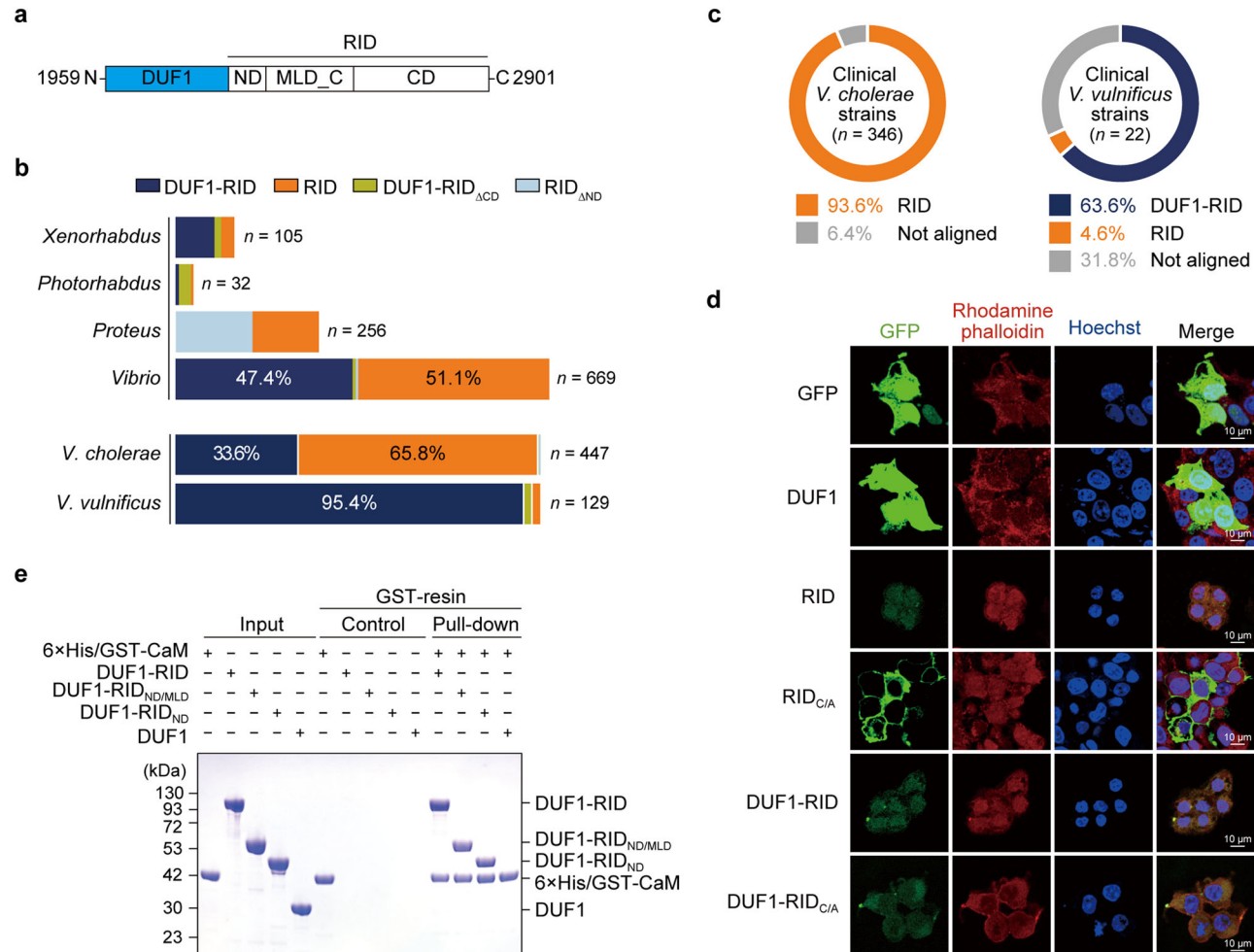

**Fig. 1 | Pathogenic bacteria use the DUF1-RID effector duet within the MARTX toxin to hijack calcium-free calmodulin during infection. a** Schematic diagram showing the domains of the DUF1-RID module within the MARTX toxin secreted by *V. vulnificus* MO6-24/O strain. DUF1 domain of unknown function in the first position, RID Rho inactivation domain, ND N-terminal domain (RID_ND), MLD_C membrane localization domain-containing domain (RID_MLD_C), CD catalytic domain (RID_CD). **b** Analysis of the DUF1-RID module, and its variants containing RID, within the MARTX toxin from different bacterial genera. **c** Characteristics of effectors within the MARTX toxin from clinical *V. cholerae* and *V. vulnificus* strains. Not aligned, not aligned with the DUF1-RID module. **d** Cytopathogenicity of the indicated proteins. GFP or GFP-fused proteins were expressed in HEK293T cells and analyzed by confocal imaging. Note that the expression level of DUF1-RID, DUF1-RID_C/A, and RID was much lower than that of RID_C/A due to their inherent cytotoxicity. **e** In vitro pull-down assay showing the interactions between DUF1-RID and its variants with calmodulin (CaM). The data shown in **d** and **e** are representative of three independent experiments, each with similar results. Source data are provided as a Source Data file.

DUF1 within the DUF1-RID_CBD structure comprises the N-terminal domain (DUF1_ND, residues 1959–2227), formed by a bundle of eight α helices (α1–α8), three $3_{10}$ helices ($3_{10}$h1–$3_{10}$h3), and four short β strands (β1–β4), as well as a small lid domain (DUF1_Lid, residues 2228–2264) formed by an α helix (α9), a $3_{10}$ helix ($3_{10}$h4), and two short β strands (β5 and β6) (Fig. 2a and Supplementary Fig. 2a). RID_CBD composes three helices (α10–α12) and a mixed parallel and antiparallel beta sheet (β7–β12) (Fig. 2a and Supplementary Fig. 2a). Of note, the two effectors are connected by α10, which forms part of the RID_CBD.

Upon interaction with CaM, DUF1-RID_CBD undergoes a marked conformational change (Fig. 2b, c). Although $Ca^{2+}$-free CaM interacts much more strongly with the effector duet than $Ca^{2+}$-bound CaM (Supplementary Fig. 1e), the structure of both CaM proteins complexed with DUF1-RID_CBD is identical, and resembles the $Ca^{2+}$-free CaM structure[24] that adopts two N- and C-lobes separated by a flexible linker (Supplementary Fig. 2e). The C-lobe (residues, 81–148) forms a hydrophobic groove and tightly grips the RID_CBD α10 (Fig. 2b, c and Supplementary Fig. 2f). The hydrophobic face of α10 comprises L2266, I2269, M2272, and L2273, which form extensive nonpolar interactions

with the C-lobe groove. The C-lobe αF and αG are sandwiched between DUF1 α1 and α8 and RID_CBD α10, and stabilize the complex structure via intensive hydrophobic and hydrophilic interactions (Fig. 2b, c and Supplementary Fig. 2g). The N-lobe (residues, 4–80) αA and αB form strong interactions with the α11, α12, and C-terminus loop of RID_CBD through multiple hydrophobic and hydrophilic contacts (Fig. 2b, c and Supplementary Fig. 2g). These interactions result in notable conformational changes, leading to a large rotation of α10 (-67.4°) and the rest region (-93.4°) of the RID_CBD (Fig. 2c). Along with the large structural movement of the RID_CBD, helices α1, α8, and α9 move toward the center of DUF1 by 8.2 Å, 8.9 Å, and 11 Å, respectively (Fig. 2c). In detail, 46 residues in DUF1-RID_CBD, and 40 residues in CaM, are involved in maintaining the structure of the complex via formation of 242 hydrophobic contacts, 25 hydrogen bonds, and eight salt bridges (Supplementary Fig. 2g, h). Collectively, these data suggest that hijacking host CaM triggers marked rearrangement of the domains within DUF1-RID_CBD.

Unexpectedly, the DUF1 moiety within the duet-CaM complex showed the highest structural similarity (Z-score = 6.9) to human CD38 (cluster of differentiation 38, PDB ID: 6VUA), known

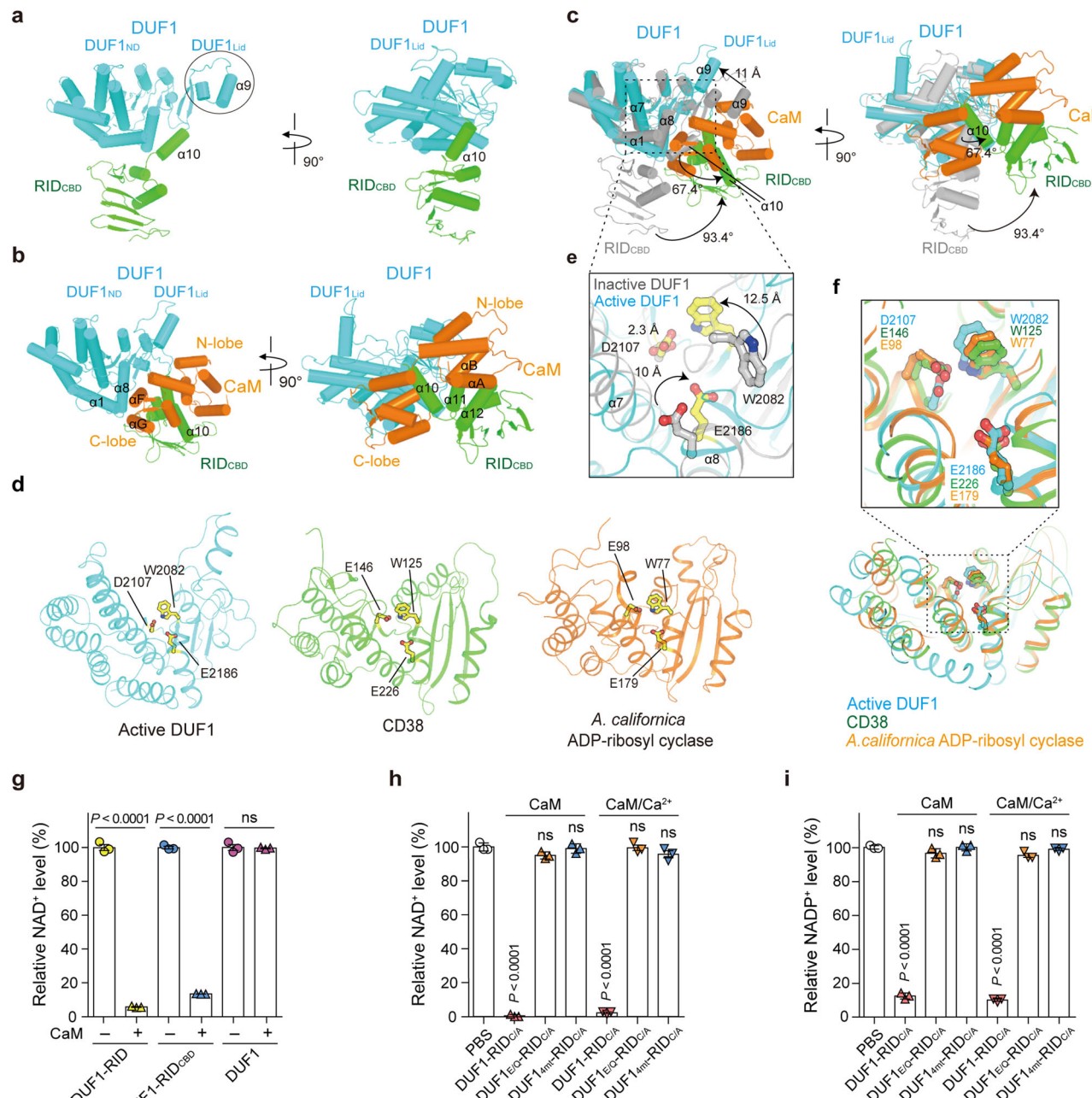

**Fig. 2 | DUF1 is a NAD(P)⁺ hydrolase that is transformed via RID-dependent CaM-binding.** Overall structures of DUF1-RID_CBD (**a**) and DUF1-RID_CBD complexed with CaM (**b**). The structures are shown as cartoon diagrams. **c** Superimposition of the DUF1-RID_CBD structure (gray) onto the CaM-binding DUF1-RID_CBD structure. The arrows indicate significant conformational changes induced in DUF1-RID_CBD upon interaction with CaM. **d** Structural comparison between DUF1 within DUF1-RID_CBD complexed with CaM, human CD38 (PDB ID 1YH3), and *A. californica* ADP-ribosyl cyclase (PDB ID: 1R12). The catalytic residues in each protein are represented by sticks. **e** Switching of non-functional catalytic residues (gray) within DUF1 to functional residues (yellow) within DUF1-RID_CBD complexed with CaM.

**f** Superimposition of the functional catalytic site of DUF1 onto that of human CD38 or *A. californica* ADP-ribosyl cyclase. Catalytic residues are displayed. **g** In vitro NAD⁺-hydrolyzing assay for DUF1-RID and its truncates, with or without CaM. Levels of NAD⁺ relative to PBS are represented ($n = 3$ per group). In vitro NAD⁺- or NADP⁺-hydrolyzing assay for DUF1-RID and its mutants complexed with CaM or Ca²⁺-bound CaM. NAD⁺ (**h**) or NADP⁺ (**i**) levels relative to PBS are represented ($n = 3$ per group). Data shown in **g**–**i** are representative of three independent experiments, each with similar results (mean ± SEM). *P*-values were calculated using one-way ANOVA with multiple comparisons. ns, not significant. Source data are provided as a Source Data file.

primarily as a nicotinamide adenine dinucleotide (NAD⁺) hydrolase (Fig. 2d). A structural comparison revealed that residues W2082, D2107, and E2186 in the duet-CaM complex, which are shifted by 12.5 Å, 2.3 Å, and 10.0 Å (Fig. 2e), respectively, upon CaM binding, adopt the same conformations as catalytically important residues W125, E146, and E226 in CD38, respectively (Fig. 2d, f). In addition to CD38, the structure of the DUF1-CaM

complex is similar to that of *Aplysia californica* ADP-ribosyl cyclase (Fig. 2d, f).

Subsequently, we assessed the NAD⁺-hydrolyzing activity of the DUF1-RID duet using purified proteins. An in vitro NAD⁺-hydrolyzing assay clearly showed that the effector duet hydrolyzed NAD⁺ in a CaM-dependent manner (Fig. 2g). The DUF1-RID_CBD, but not DUF1 alone, showed NAD⁺-hydrolyzing activity comparable with that of the effector

duet, indicating that the RID_CBD is critical for the enzymatic function of DUF1 via interaction with CaM, as shown by the structure of the complex (Fig. 2g).

We then used DUF1-RID_C/A containing an inactive RID to further validate the enzymatic activity of DUF1. The effector duet showed full NAD+-degrading activity, indicating that enzymatic activity is independent of the RID function that inactivates Rho GTPases (Fig. 2h). Substitution of the active residue E2186 in DUF1 (corresponding to the active residue E226 in CD38) with glutamine (DUF1_E/Q-RID_C/A) completely abolished the NAD+-degrading activity (Fig. 2h). The DUF1-RID_C/A was furthermore mutated at residues critically important for the interaction with CaM (i.e., W2183L, R2195L, E2285L, and R2328L), thereby generating DUF1_4mt-RID_C/A (Supplementary Fig. 2i). This mutant also lacked NAD+-hydrolyzing activity (Fig. 2h). Since CD38 also has NADP+-hydrolyzing activity[25], we tested whether DUF1 has the same activity and found that DUF1 does indeed hydrolyze NADP+ (Fig. 2i). Consistent with the biochemical and structural analysis, these data suggest that the enzymatic activity of CaM-mediated DUF1 is independent of calcium (Fig. 2h, i).

Next, we used nuclear magnetic resonance (NMR) spectroscopy and liquid chromatography-tandem mass spectrometry (LC-MS/MS) to examine the by-products generated in vitro by DUF1-RID_C/A-mediated hydrolysis of NAD(P)+. The results showed that DUF1-RID_C/A hydrolyzes the NAD+ to yield nicotinamide (NAM) and ADP-ribose (ADPR) (Supplementary Fig. 4a–c). In addition, the effector duet hydrolyzed NADP+ to yield NAM and ADP-ribose 2'-phosphate (Supplementary Fig. 4c). Taken together, these data demonstrate that DUF1 is transformed into a NAD(P)+-hydrolyzing enzyme by RID-dependent

CaM-binding. Thus, we renamed DUF1 the "RID dependently transforming NADase domain (RDTND)".

## Molecular details of RDTND-RID_CBD/CaM complexed with NAD+

While CD38 is responsible for >99% of NAD+ hydrolysis to generate ADPR and NAM, it also functions as a very inefficient ADP-ribosyl cyclase to synthesize cyclic ADPR (cADPR) from NAD+ (cADPR comprises less than 1% of reaction products)[26–30]. To provide structural insight into the enzymatic activity and potential substrate promiscuity of RDTND, we next determined the structure of enzymatically inactive RDTND_E/Q-RID_CBD/CaM in complex with NAD+ (Fig. 3a and Supplementary Table 1). Overall, the NAD+-bound structure is similar to the RDTND-RID_CBD/CaM structure (root mean square deviation of 0.97 Å); however, the RDTND_Lid shows marked movement when bound to NAD+ (Fig. 3b). In detail, the adenine group of NAD+ forms a cation-π interaction with R2234, hydrogen bonds with G2179, and hydrophobic contacts with F2001, F2150, and Y2181 (Fig. 3a and Supplementary Fig. 5). In addition, the 2"-hydroxyl group in the 2R-ribose moiety interacts with (Fig. 3a and Supplementary Fig. 5). These interactions rotate the α9 helix by 42°, bringing RDTND_Lid closer to the catalytic site (Fig. 3b), suggesting that RDTND_Lid may play a role in recruiting substrate. The di-phosphate group of NAD+ forms hydrogen bonds with S2083, G2084, R2177, N2180, Y2181, and F2182 (Supplementary Fig. 5). The 1R-ribose group forms a hydrophobic contact with W2082, and hydrogen bonds with D2107 and E2186Q (Supplementary Fig. 5). The nicotinamide group interacts with W2082, I2117, T2153, and Y2181 via hydrophobic contacts, and with D2107 through hydrogen bonding (Supplementary Fig. 5). In particular, residues W2082, D2107, and

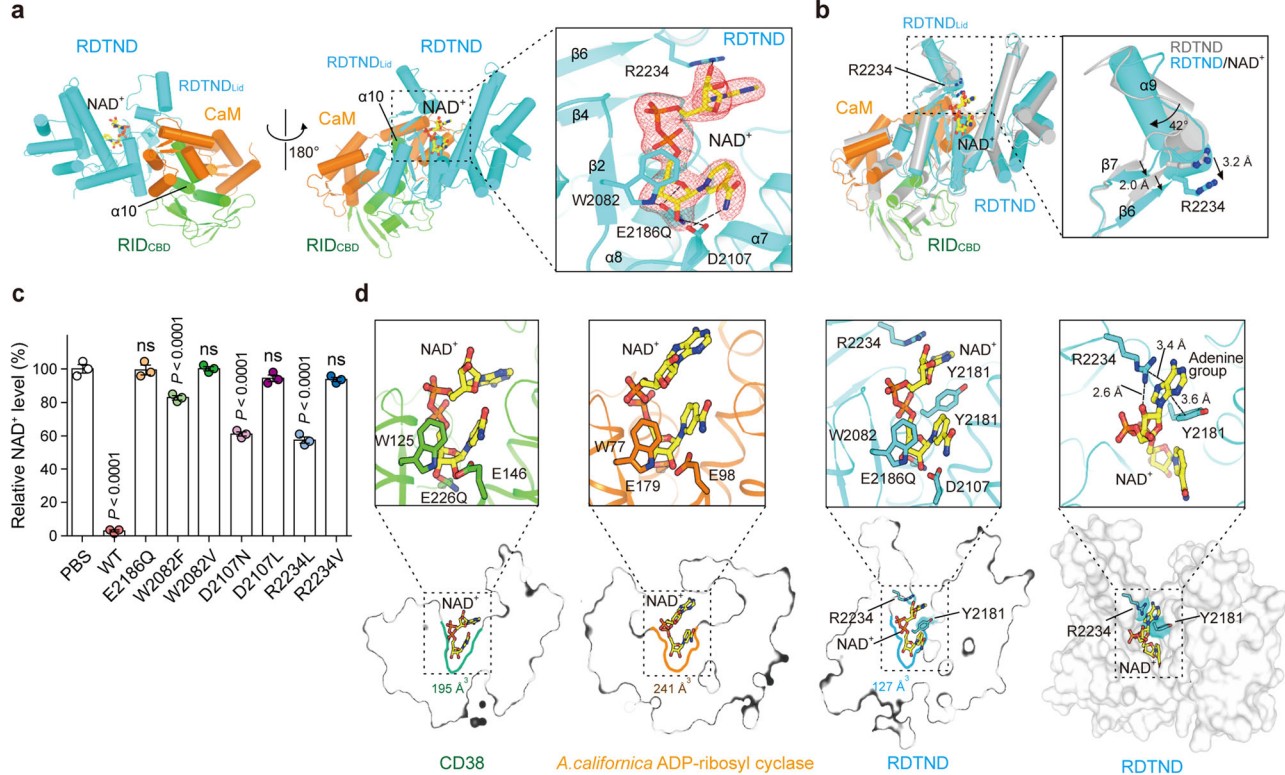

**Fig. 3 | Detailed molecular structure of RDTND-RID_CBD/CaM/Ca2+ complexed with NAD+. a** Overall structure of RDTND-RID_CBD/CaM complexed with NAD+, and the binding mode of NAD+ at the catalytic site (magnified panel). The mFo-DFc omit electron density map of NAD+ contoured at 3.0 σ is shown as a mesh (red). **b** Superimposition of the NAD+-bound RDTND-RID_CBD/CaM structure onto the RDTND-RID_CBD/CaM structure (gray). Significant conformational changes in RDTND_Lid are shown (magnified panel). **c** In vitro NAD+-hydrolyzing assay for CaM-bound RDTND-RID_C/A and its RDTND mutants. The NAD+ levels relative to the PBS

control are presented (*n* = 3 per group). Data are representative of three independent experiments, each with similar results (mean ± SEM). *P*-values were calculated using one-way ANOVA with multiple comparisons. ns, not significant. **d** Structural comparison of active NAD+-bound RDTND with that of NAD+-bound human CD38 (PDB ID 2I65) and NAD+-bound *A. californica* ADP-ribosyl cyclase (PDB ID: 3ZWM). Structures are shown as cartoon (upper panel) or surface diagrams (lower panel). Source data are provided as a Source Data file.

E2186Q are critically involved in maintaining the stable conformation of the nicotinamide and 1R-ribose sugar (Fig. 3a). These structural data suggested that W2082, D2107, E2186, and R2234 are critical for substrate interaction and/or catalytic activity. Data from the in vitro $NAD^+$-hydrolyzing assay clearly show that mutations at these residues completely abolish the enzymatic function of RDTND (Fig. 3c).

To assess the possibility that RDTND plays a role in the synthesis of cADPR from $NAD^+$, we compared the structures of RDTND-$RID_{CBD}$/CaM/$NAD^+$, CD38/$NAD^+$, and *Aplysia californica* ADP-ribosyl cyclase/$NAD^+$ (Fig. 3d). The mode by which RDTND interacts with $NAD^+$ is similar to the way in which CD38 and *Aplysia californica* ADP-ribosyl cyclase interact with $NAD^+$ at its catalytic site; however, the $NAD^+$ binding pocket within the catalytic domain is significantly different between the three enzymes. Unlike CD38 and *Aplysia californica* ADP-ribosyl cyclase, in which the catalytic site pocket has no spatial restriction with respect to cyclization of ADPR after hydrolysis of $NAD^+$, RDTND has a smaller and narrower pocket due to the bulky residues R2234 and Y2181. In particular, Y2181 is positioned between the adenine and NAM moieties of $NAD^+$, and therefore obstructs rotation of the adenine ring during the ADPR cyclization reaction (Fig. 3d).

We further evaluated the ADP-ribosyl cyclase activity of RDTND complexed with RID/CaM by using a modified HPLC-UV method[31]. Consistent with previously reported results[26,28,29], the analyses showed that CD38 mostly hydrolyzes $NAD^+$ to ADPR (99.28%) and generates a small amount of cADPR (0.74%) (Supplementary Fig. 6). RDTND complexed with RID/CaM hydrolyzed $NAD^+$ to ADPR (99.69%) and cADPR (0.31%). The structural restriction on the movement of $NAD^+$ moieties may lead to the RDTND-RID/CaM complex having lower ADP-ribosyl cyclase activity than that of CD38. Thus, the structural features together with the biochemical analysis suggest that RDTND is a $NAD(P)^+$-hydrolyzing enzyme, although it cannot be ruled out that it has ADP-ribosyl cyclase activity.

## Cryo-electron microscopic (EM) structure of the RDTND-RID complexed with CaM and Rac1

To clarify a mutual dependency of RDTND and RID during interaction with host targets, we conducted cryo-EM studies. Size-exclusion chromatography assays clearly showed that RDTND-$RID_{C/A}$ interacts with both CaM and the active form of Rac1 (i.e., $Rac1_{Q61L}$), and forms a complex with them (Supplementary Fig. 7a). Consistent with previous results[14], the duet did not interact with the inactive form of Rac1 (i.e., $Rac1_{T17N}$) but bound with CaM, suggesting critical involvement of the Switch I region of Rac1 in the interaction with RID, and the independence of CaM and Rac1 during the interaction with the effector duet (Supplementary Fig. 7b). Subsequently, we obtained a complex comprising RDTND-$RID_{C/A}$, CaM, and $Rac1_{Q61L}$ (residues 1–177) through a sequence of purification procedures, including chemical cross-linking to stabilize the complex (Supplementary Fig. 8). Note that we deleted the C-terminal PBR of Rac1 to prevent non-specific cross-linking between lysine residues within the PBR. Finally, single particle analysis enables us to generate a reconstructed 3D map of the complex at an average resolution of 4.32 Å (Supplementary Figs. 8 and 9, and Supplementary Table 2). We initially tried to determine the cryo-EM structure of the non-cross-linked complex under various conditions including the use of grids and vitrification. However, single particle cryo-EM image analysis typically resulted in low-resolution 2D class averages, indicating flexible movement within the complex (Supplementary Fig. 10), caused possibly by low binding affinity between RID and Rac1. Cross-linking reduced flexible movement in the sample and improved the overall resolution of the cryo-EM map. The cryo-EM density map of the RDTND-$RID_{CBD}$ complexed with CaM is observed at a contour of 8–12 σ, while the regions of $RID_{C/A}$ complexed with $Rac1_{Q61L}$ are relatively weak in the cryo-EM density map contoured at ~4.5–6.0 σ (Supplementary Fig. 11); these data correlate with the

binding affinity between $MARTX_{Vv}$ RDTND-$RID_{C/A}$ and CaM (Supplementary Fig. 1e, $K_D$ = 80 nM), and between $MARTX_{Vv}$ RID and Rac1 ($K_D$ = 20 μM)[14]. Consequently, each protein structure in the complex fitted well into the cryo-EM density map contoured at 4.5 σ (Supplementary Fig. 11).

The overall shape and dimensions of the cryo-EM map show good agreement with the crystal structures of RDTND-$RID_{CBD}$ complexed with CaM (this study), and $RID_{C/A}$ (PDB ID, 5XN7)[14] (Fig. 4a, b and Supplementary Fig. 11a–d). An extra density map was observed near to $RID_{CD}$, which fitted well with the crystal structure of the active form of Rac1 (PDB ID 3TH5) (Fig. 4a, b and Supplementary Fig. 11e, f). The cryo-EM map indicated that Switch I is critical for the interaction with $RID_{CD}$ (Supplementary Fig. 11e). These structural conformations reveal that the PBR of Rac1 is accessible to the catalytic site of RID (Fig. 4b). The complex forms an L-shaped structure (Fig. 4a, b). Consistent with the biochemical analyses (Supplementary Fig. 7), these data suggest that CaM and Rac1 are associated independently with the effector duet within the complex.

RID functions as an $N^\epsilon$-fatty acyltransferase that modifies lysine residues within the PBR of Rac1 at the plasma membrane[14]. The cryo-EM structure reveals a positively charged surface spanning the entire RID, except for $RID_{CBD}$ (Fig. 4c). To elucidate the cellular localization of the effector module, we expressed the enzymatically inactive $RDTND_{E/Q}$-$RID_{C/A}$ duet fused with the C-terminal green fluorescent protein (GFP), as well as its truncated proteins, ectopically in HeLa cells. A green fluorescence signal was emitted by the effector duet exclusively at the plasma membrane, suggesting that the duet re-localizes to the plasma membrane after CPD-mediated release from the MARTX toxin (Fig. 4d). A previous study used a protein-lipid overlay assay to reveal a specific interaction between the $MARTX_{Vv}$ RID and PI(4,5)P2[14]. That study suggested that a positively charged pocket in the RID N2 subdomain (corresponding to the $RID_{MLD\_C}$ in this study) may bind to PI(4,5)P2. In the present study, further domain analysis revealed that the membrane localization of the duet is dependent on the entire RID, including both $RID_{MLD\_C}$ and $RID_{CD}$ (Fig. 4d). The truncated forms of RDTND-$RID_{CBD/MLD\_C}$ (without $RID_{CD}$) or RDTND-$RID_{CBD/MLD}$ are insufficient for localization to the plasma membrane. The WT RDTND lacking RID localized exclusively to the cytoplasm and did not induce any morphological changes in cells, further indicating the RID-dependent function of RDTND (Fig. 4d).

Upon discharge from the MARTX toxin, the positively charged surface of RID mediates re-localization to the membrane, and this conformation localizes RDTND exclusively in the cytoplasm (Fig. 4e, upper). $RID_{CBD}$-dependent CaM binding transforms RDTND into a functional enzyme that seeks out $NAD(P)^+$, located either in the cytoplasm or near to the membrane (Fig. 4e, lower). The structural conformation also enables RID to interact with Rac1 located at the plasma membrane via interaction between the lysine residues in the PBR of the substrate and the active site (Fig. 4e, lower). It should be mentioned however that the interaction between the $MARTX_{Vv}$ RID and Rac1 ($K_D$ = 20 μM) is significantly weaker than that between the $MARTX_{Vc}$ RID and Rac1 ($K_D$ = 4 μM)[14]. $MARTX_{Vv}$ RID (residues 2264–2901, NCBI accession ID: WP_015728045.1) and $MARTX_{Vc}$ RID (residues 2465–3098, NCBI accession ID: WP_108347803.1) share a high level of sequence identity (85.6%) and similarity (93.5%) (Supplementary Fig. 12a). We used AlphaFold 3[32] to generate a structural model of the $MARTX_{Vc}$ RID using the template structure of $MARTX_{Vv}$ RID[14]. We compared positively charged residues on the surfaces of $MARTX_{Vv}$ RID, $MARTX_{Vv}$ RID within the cryo-EM complex structure, and the modeled $MARTX_{Vc}$ RID (Supplementary Fig. 12b). Positive residues are distributed similarly on the surfaces of the RID proteins (Supplementary Fig. 12c). More positive residues occur on the surface of $MARTX_{Vc}$ RID than on the surface of $MARTX_{Vv}$ RID, particularly in the membrane localization domain (MLD)-containing domain (i.e., MLD_C), suggesting that membrane binding of $MARTX_{Vc}$ RID may be stronger than that

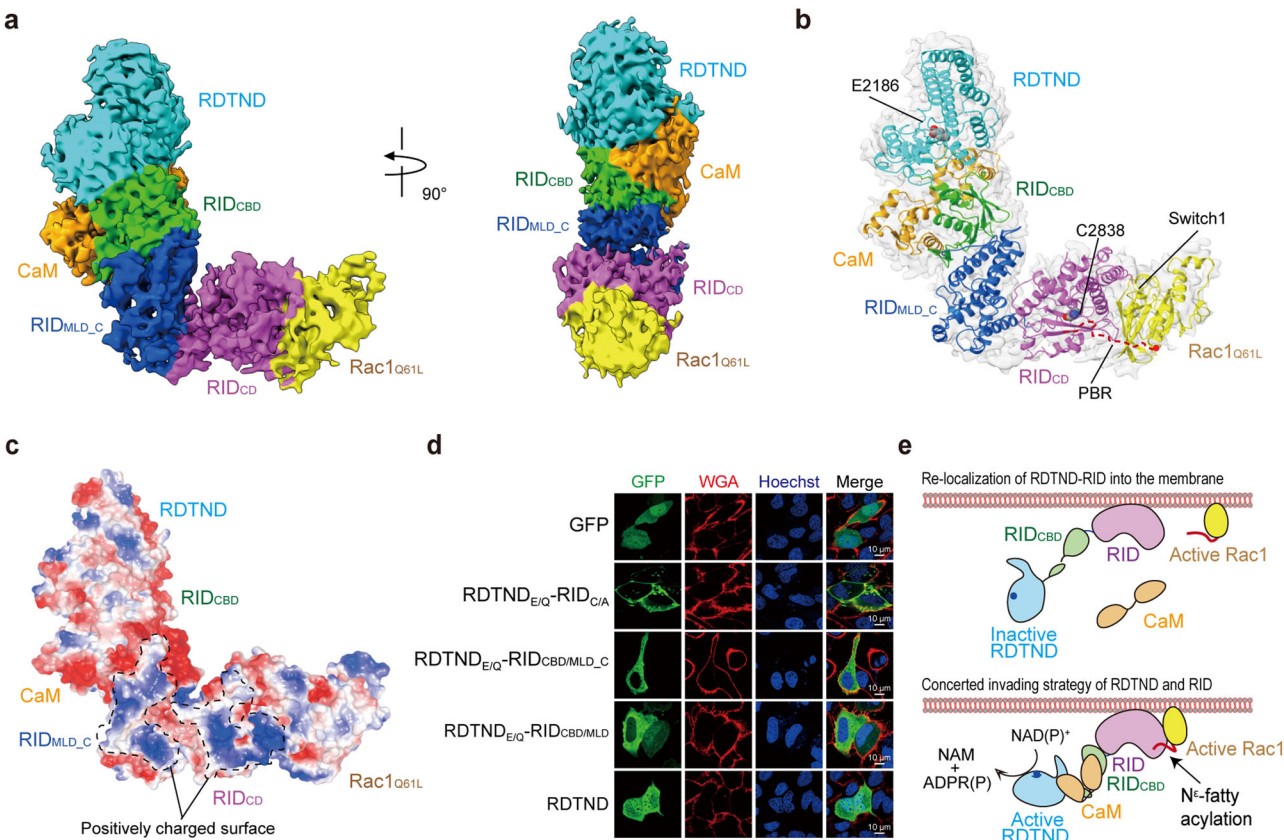

**Fig. 4 | Cryo-EM structure of RDTND-RID$_{C/A}$ complexed with CaM and Rac1$_{Q61L}$.** Reconstructed 3D cryo-EM map (**a**) and the fitted model (cartoon representation) of RDTND-RID$_{C/A}$ complexed with CaM and Rac1$_{Q61L}$ (**b**). Catalytic residue C2838 in RID, and the C-terminal polybasic (PBR, red dashed line) and Switch 1 regions in Rac1$_{Q61L}$, are indicated. **c** Surface charge distribution of the RDTND-RID$_{C/A}$/CaM/ Rac1$_{Q61L}$ complex. The negative and positive charges are shown as red and blue, respectively. The positively charged surface of RID$_{MLD\_C}$ and RID$_{CD}$ is highlighted by a dashed line. **d** Subcellular localization of the RDTND-RID module and its truncates. The GFP, GFP-fused RDTND$_{E/Q}$-RID$_{C/A}$, or GFP-fused RDTND-RID truncates (RDTND$_{E/Q}$-RID$_{CBD/MLD\_C}$, RDTND$_{E/Q}$-RID$_{CBD/MLD}$, and RDTND) were expressed in HeLa cells and analyzed. The HeLa cell plasma membrane and nucleus were stained with CF® 594-conjugated wheat germ agglutinin (WGA) and Hoechst 33342, respectively. Data are representative of three independent experiments, each with similar results. **e** Cartoon summarizing the re-localization of RDTND-RID to the membrane after CPD-mediated release from the MARTX toxin, as well as the concerted invasion strategy used by the effector duet at the membrane and near the membrane.

of MARTX$_{Vv}$ RID. The weaker membrane binding of MARTX$_{Vv}$ RID may be due the structural flexibility needed to transform RDTND into a NAD(P)$^+$-hydrolyzing enzyme via hijacking of CaM, which may be more associated with the weaker interaction between MARTX$_{Vv}$ RID and Rac1 than the interaction between MARTX$_{Vc}$ RID and Rac1. The binding affinities of MARTX$_{Vc}$ RID to Rac1 and MARTX$_{Vv}$ RID to Rac1 were calculated in solution[14]. Since RID proteins are localized to the membrane and RID-catalyzed Rac1 modification is achieved in the membrane[14], the binding affinities may be higher between both RID proteins and Rac1 in the membrane. Thus, the architectural organization of the duet in complex with host targets provides important insights into the concerted invasion strategy mediated by the MARTX toxin.

## RDTND-RID silences the ROS/ERK/JNK/NF-κB axis critically required for first line immune responses in the host

Upon bacterial infection, host immune cells such as macrophages and neutrophils rapidly generate ROS to clear bacteria through NOX family proteins sited either at the plasma membrane or within mitochondria[33–38]. ROS activate MAPK signaling, which in turn triggers pro-inflammatory immune responses[39,40]. Therefore, eliminating ROS allows bacteria to survive and colonize the host[41,42]. Since depletion of NAD$^+$ from immune cells causes a failure of cellular redox homeostasis, which is critical for ROS production via NOX family proteins[43], and small GTPase Rac1 or Rac2 acts as a regulatory subunit during

activation of various NOX family members[15], we evaluated the significance of the effector duet in bone marrow-derived macrophages (BMDMs) infected with a series of engineered *V. vulnificus* strains that secrete MARTX toxins carrying effector-free (EF), effector duet (EF::RDTND-RID), or mutant duet (EF::RDTND-RID$_{C/A}$, EF::RDTND$_{E/Q}$-RID$_{C/A}$, or EF::RDTND$_{4mt}$-RID$_{C/A}$) (Fig. 5a).

First, we evaluated intracellular NAD$^+$ and NADP$^+$ levels, both of which were significantly decreased in BMDMs infected with *V. vulnificus* strains secreting MARTX toxins containing EF::RDTND-RID or EF::RDTND-RID$_{C/A}$, but not in BMDMs infected by other strains containing inactive RDTND (EF::RDTND$_{E/Q}$-RID$_{C/A}$) or an effector duet defective in CaM binding (EF::RDTND$_{4mt}$-RID$_{C/A}$). These data further demonstrate the function of RDTND as an NAD(P)$^+$-hydrolyzing enzyme at the strain level (Fig. 5b, c).

Next, we used flow cytometry to analyze intracellular ROS levels in RAW 264.7 macrophage cells infected with different strains (Fig. 5d, e, and Supplementary Fig. 13). The intracellular ROS level increased upon infection with EF or inactive RDTND-containing toxin strains (EF::RDTND$_{E/Q}$-RID$_{C/A}$) due to LPS-mediated ROS generation[33,44]; however, infection by the EF::RDTND-RID$_{C/A}$ strain reduced ROS production by macrophages significantly. We were unable to calculate ROS production induced by the *V. vulnificus* strain secreting a MARTX toxin carrying EF::RDTND-RID due to severe cytotoxic effects on immune cells. Nevertheless, a previous study showed that MARTX$_{Vc}$ RID inhibits

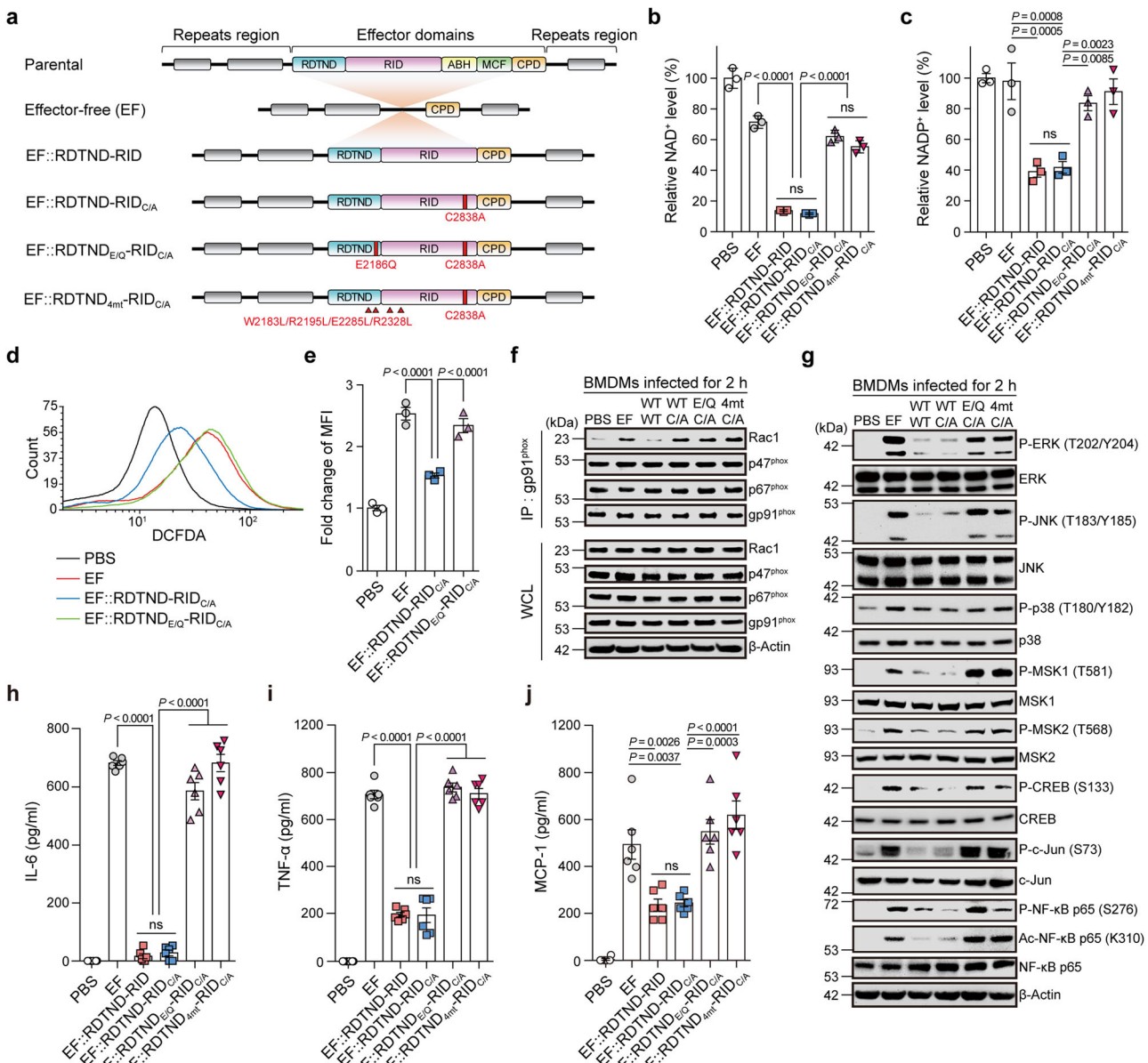

**Fig. 5 | The RDTND-RID duet silences the ROS/ERK/JNK/NF-κB axis, which is critical for first line immune responses. a** Schematic diagram showing MARTX toxins from engineered *V. vulnificus* mutant strains. Substituted residues in RDTND-RID within the MARTX toxin from each mutant strain are indicated. Intracellular NAD[+] (**b**) and NADP[+] (**c**) levels in BMDMs infected for 2 h with the indicated *V. vulnificus* strains (MOI = 5, 5 × 10[6] CFU). NAD(P)[+] levels relative to PBS are presented (*n* = 3 per group). ROS level in RAW 264.7 cells infected for 2 h with the indicated *V. vulnificus* strains (MOI = 5, 5 × 10[6] CFU). ROS production was measured by flow cytometry (**d**), and fold changes in mean fluorescence intensity (MFI) compared with the PBS control are shown (*n* = 3 per group) (**e**). **f** Immunoprecipitation assay. Lysates of BMDMs infected for 2 h with the indicated *V. vulnificus* strains were immunoprecipitated by an anti-gp91[phox] antibody, followed by immunoblot analysis with anti-Rac1, anti-p47[phox], anti-p67[phox], and anti-gp91[phox] antibodies. **g** Immunoblot analysis of MAPK signaling molecules and transcription factors in BMDMs infected for 2 h with the indicated *V. vulnificus* strains. EF, effector-free; WT-WT, EF::RDTND-RID; WT-C/A, EF::RDTND-RID_{C/A}; E/Q-C/A, EF::RDTND_{E/Q}-RID_{C/A}; 4mt-C/A, EF::RDTND_{4mt}-RID_{C/A}. ELISA of pro-inflammatory cytokines IL-6 (**h**), TNF-α (**i**), and MCP-1 (**j**) in supernatants from BMDMs infected for 2 h with the indicated *V. vulnificus* strains (*n* = 6 per group). Data shown are representative of three independent experiments, each with similar results (mean ± SEM). *P*-values were calculated using one-way ANOVA with multiple comparisons. ns, not significant. Source data are provided as a Source Data file.

ROS production by mouse macrophage cells[14]. We then examined involvement of Rac1 in formation of NOX2 complexes to defend against infection. Immunoprecipitation (IP) assays showed that infection by the EF, EF::RDTND-RID_{C/A}, EF::RDTND_{E/Q}-RID_{C/A}, or EF::RDTND_{4mt}-RID_{C/A} led to a marked increase in the NOX2/gp91[phox] interaction with Rac1, p47[phox], and p67[phox] in macrophages (Fig. 5f). However, infection by the EF::RDTND-RID strain infection led to a significant reduction in the association between the key component Rac1 and these subunits (Fig. 5f), resulting in NOX2 inactivation and

inhibition of ROS generation. The cryo-EM structure reveals that RID interacts with active Rac1 via the Switch I region and allows Rac1 to access the PBR for fatty acylation modification (Fig. 4b and Supplementary Fig. 11e). A previous structural and interaction study using purified proteins expressed in *Escherichia coli* showed that active Rac1 interacts with p67[phox] (*K*_D = 1.7 μM) via the Switch I region[45]. Rac1 contains two potent membrane-localizing elements (the PBR and a geranylgeranyl modification) at its C-terminus[46]. The transmembrane gp91[phox] catalytic subunit (i.e., NOX2) is activated by the assembly of

four cytosolic subunits (p47$^{phox}$, p67$^{phox}$, Rac1 or Rac2, and p40$^{phox}$), which translocate to the membrane to form the active NOX2 complex. In preparation for membrane translocation, p67$^{phox}$ interacts with p47$^{phox}$ prior to the interaction with the GTP-bound Rac protein[47]. Although the binding affinity between MARTX$_{Vv}$ RID and Rac1 ($K_D = 20$ μM)[14] is weaker than that between p67$^{phox}$ and Rac1, the IP results (Fig. 5f) suggest that the association between modified Rac1 and the p67$^{phox}$ and p47$^{phox}$ complex may reduce the binding affinity, and that RID competitively hijacks and fatty acylates Rac1 to block its association with the NOX2 complex in macrophages infected by the EF::RDTND-RID strain. Collectively, these data suggest that both NAD(P)$^+$ depletion and Rho GTPase modification by the RDTND-RID duet are critically linked to reduced production of ROS.

During early responses, ROS act as secondary messengers capable of regulating expression of pro-inflammatory genes by altering kinase cascades and activating transcription factors[48–52]. Therefore, we examined MAPK cascades in BMDMs infected with *V. vulnificus* strains at different time points. We found no significant difference in cellular levels of phosphorylated extracellular signal-regulated kinase (ERK) or c-Jun NH2-terminal kinase (JNK) between BMDMs infected with EF, EF::RDTND-RID$_{C/A}$, or EF::RDTND$_{E/Q}$-RID$_{C/A}$ up until 1 h post-infection. However, there was a significant reduction in phosphorylation of ERK and JNK in immune cells infected with strain EF::RDTND-RID$_{C/A}$, but not with strain EF or EF::RDTND$_{E/Q}$-RID$_{C/A}$, at 2 h post-infection (Supplementary Fig. 14a). Thus, we further analyzed the signaling pathways at 2 h post-infection, and observed that activation of ERK and JNK was reduced markedly in cells infected with EF::RDTND-RID or EF::RDTND-RID$_{C/A}$ (Fig. 5g). However, activation of p38 was unchanged in cells infected with these two strains when compared with strains EF, EF::RDTND$_{E/Q}$-RID$_{C/A}$, or EF::RDTND$_{4mt}$-RID$_{C/A}$ (Fig. 5g).

Mitogen- and stress-activated kinases (MSK) are activated by either ERK or p38 kinases in response to stress, including bacterial infection; activated ERK or p38 kinases subsequently activate pro-inflammatory mediators such as NF-κB in monocyte/macrophage cells[44,53,54]. The silenced phosphorylation of ERK and JNK mediated by RDTND-RID resulted in reduced phosphorylation of MSK1/2, which was linked to a significant reduction in phosphorylation of CREB (S133) and NF-κB (S276) (Fig. 5g). In addition to phosphorylation of the NF-κB p65 subunit at S276, acetylation at K310 is required for NF-κB-mediated transactivation of pro-inflammatory genes. We found that effector duet-mediated suppression of MAPK signaling also silenced acetylation of NF-κB p65 at K310 significantly (Fig. 5g). By contrast, reduced phosphorylation of JNK inhibited activation of c-Jun, a critical regulator of pro-inflammatory cytokine expression in macrophages (Fig. 5g).

Next, we evaluated the expression levels of pro-inflammatory cytokines in BMDMs infected with *V. vulnificus* strains. While BMDMs infected with EF or inactive RDTND-RID duet-containing MARTX toxin-secreting *V. vulnificus* strains produced significant levels of IL-6, TNF-α, and MCP-1, cells infected with the enzymatically active effector duet-containing toxin-secreting strains (i.e., EF::RDTND-RID strain harboring both NAD(P)ase and Nε-fatty acyl-transferase functions and EF::RDTND-RID$_{C/A}$ strain harboring only NAD(P)ase function) showed a clear reduction in pro-inflammatory cytokine production (Fig. 5h–j). We further validated the function of RDTND alone in pro-inflammatory cytokine suppression using NOX2-silenced macrophage cells infected with *V. vulnificus* strains. Overall cytokine levels in the NOX2-silenced macrophage cells infected with the strains were significantly lower than those in WT macrophage cells. Nevertheless, the results showed that only NAD(P)ase activity (i.e., EF::RDTND-RID$_{C/A}$) significantly suppressed pro-inflammatory cytokine production in the NOX2-silenced macrophage cells (Supplementary Fig. 14b–e), indicating that depletion of NAD(P)$^+$ by RDTND is critically involved in pro-inflammatory cytokine suppression via ROS reduction, independent of RID-mediated inhibition of NOX2 complex formation. Thus, these data

demonstrated that MARTX toxin-secreting pathogenic bacteria uses the RDTND-RID duet-mediated strategy to suppress efficiently ROS generation in immune cells, which silences the ERK/JNK/NF-κB signaling required for the evasion of the pro-inflammatory immune responses critical for the first line of defense against infection.

## RDTND-RID promotes bacterial dissemination and sepsis in mice

To demonstrate the pathogenicity of the effector duet in vivo, mice were subcutaneously challenged with a series of *V. vulnificus* strains. First, we assessed NAD$^+$ levels in peripheral blood mononuclear cells (PBMCs) isolated from the blood of mice at 7 h post-infection with the indicated *V. vulnificus* strains. The NAD$^+$ levels in PBMCs from mice infected with EF::RDTND-RID or EF::RDTND-RID$_{C/A}$ strains were significantly lower than those in mice infected with the EF::RDTND$_{E/Q}$-RID$_{C/A}$ or EF::RDTND$_{4mt}$-RID$_{C/A}$ strains (Fig. 6a). Next, we measured bacterial colonization of the blood, spleen, and liver in mice. The largest numbers of bacteria were found in the blood, spleen, and liver of the mice challenged with the EF::RDTND-RID strain, with slightly lower numbers in those challenged with the EF::RDTND-RID$_{C/A}$ strain (Fig. 6b–d). However, mice infected with the EF::RDTND$_{E/Q}$-RID$_{C/A}$ or EF::RDTND$_{4mt}$-RID$_{C/A}$ strains had significantly lower bacterial populations in the blood, spleen, and liver (Fig. 6b–d).

Consistent with the colonization results showing bacterial dissemination into different tissues in mice, we found that IL-6 and MCP-1 levels in sera were also considerably higher (i.e., a cytokine storm), and body temperature was substantially lower, in mice infected with the EF::RDTND-RID or EF::RDTND-RID$_{C/A}$ strain (particularly at 5 h and 7 h post-infection); these are typical signs and symptoms of bacterial sepsis (Fig. 6e–g). Furthermore, significant infiltration of the liver by immune cells was observed in mice infected with the EF::RDTND-RID or EF::RDTND-RID$_{C/A}$ strains at 10 h post-infection (Supplementary Fig. 14f).

Consequently, all mice challenged with the parental WT or EF::RDTND-RID strains succumbed to sepsis within 20 h post-infection (Fig. 6h). Mice challenged with the EF::RDTND-RID$_{C/A}$ strain showed a 60% survival rate, while all mice challenged with EF::RDTND$_{E/Q}$-RID$_{C/A}$ or EF::RDTND$_{4mt}$-RID$_{C/A}$ strain survived until the end of the experiment (Fig. 6h).

A previous study using mice infected with *V. vulnificus* secreting MARTX toxin showed that the bacterial population increased time-dependently at the dorsally located subcutaneously infected site[55]. The tissue surrounding the infected area revealed disintegrated myofibrils and necrotic adipocytes. Proliferation and dissemination of the bacteria to internal organs through the bloodstream was observed at 9 h post-infection[55]. *V. vulnificus* lacking the MARTX toxin was defective in conferring cytotoxicity to immune cells and was sensitive to phagocytosis by immune cells, as evidenced by the subsequent reduction in colonization and dissemination of the bacteria to internal organs[55]. Consistent with a previous study[55], our study also found that pro-inflammatory cytokine levels were up-regulated time-dependently (Fig. 6e, f), and that bacterial loads appeared in the blood, spleen, and liver at 10 h post-infection (Fig. 6b–d), suggesting that the subcutaneously infected *V. vulnificus* RDTND-RID strain or RDTND-RID$_{C/A}$ strain may colonize and subsequently spread to distal organs through the bloodstream. Taken together, these data demonstrate that MARTX toxin-secreting pathogenic bacteria utilize the RDTND-RID duet, which acts cooperatively to evade ROS-mediated pro-inflammatory immune responses and to promote systemic dissemination, ultimately resulting in bacterial sepsis.

## Discussion

Due to its essential role in maintaining cellular homeostasis, NAD$^+$ is a key target of pathogens[43,56–61]. Pathogenic bacteria secrete NAD$^+$-utilizing toxins capable of damaging hosts either by modulating the

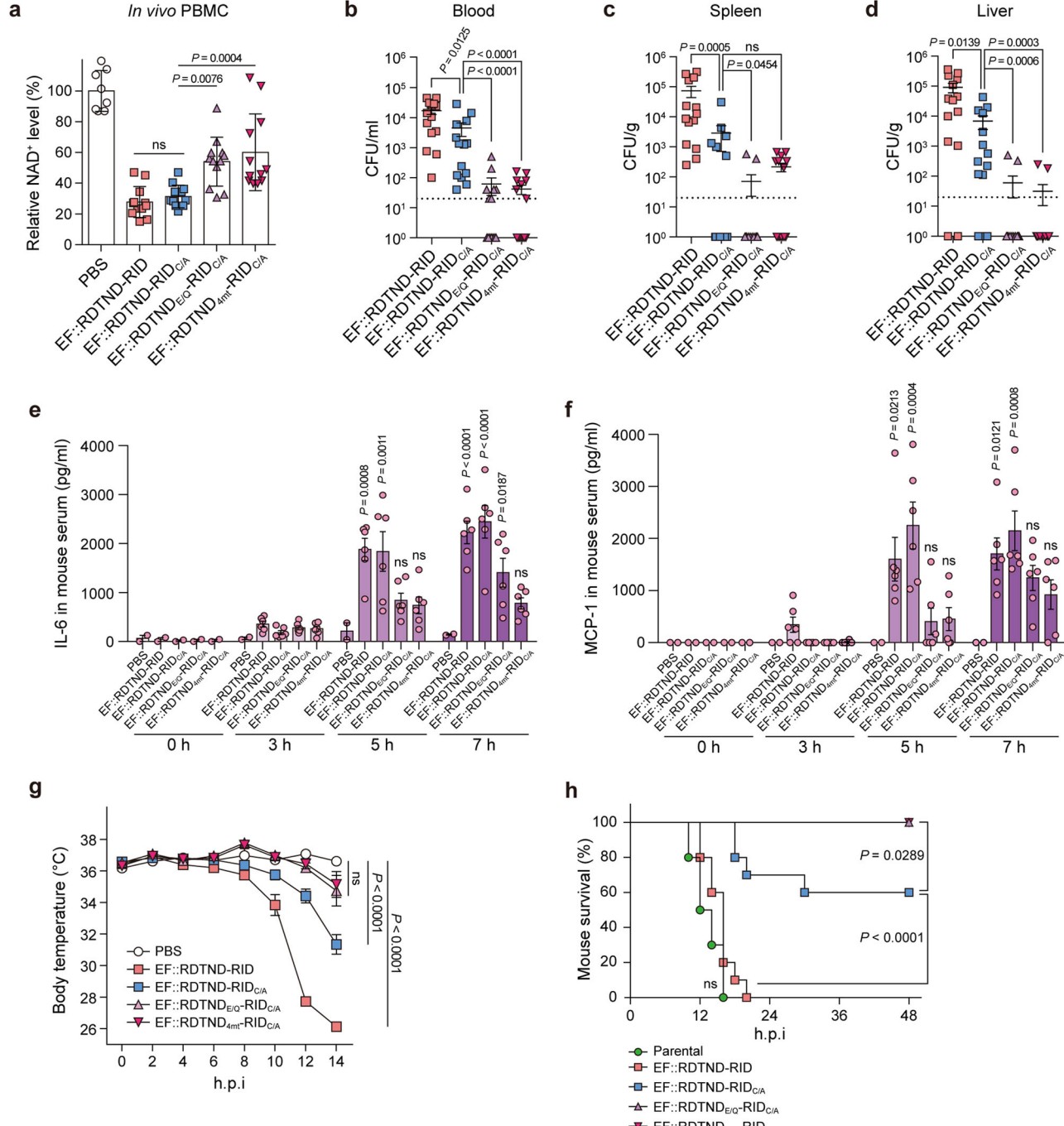

**Fig. 6 | The RDTND-RID duet promotes bacterial dissemination and sepsis in mice. a** Intracellular NAD⁺ levels in PBMCs isolated from 5-week-old female ICR mice 7 h post-infection with the indicated *V. vulnificus* strains. NAD⁺ levels relative to PBS are presented (*n* = 8 for PBS; *n* = 12 for EF::RDTND-RID, EF::RDTND-RID$_{C/A}$, EF::RDTND$_{E/Q}$-RID$_{C/A}$, and EF::RDTND$_{4mt}$-RID$_{C/A}$). Single bacterial colonies in blood (**b**), spleen (**c**), and liver (**d**) collected from 5-week-old female ICR mice at 10 h post-infection with the indicated *V. vulnificus* strains; normalized data are shown (*n* = 14 per group). ELISA of pro-inflammatory cytokines of IL-6 (**e**) and MCP-1 (**f**) in serum collected from 5-week-old female ICR mice infected with the indicated *V. vulnificus* strains at the designated time (*n* = 2 for 0 h, and *n* = 6 for 3 h, 5 h, and 7 h post-infection per group). *P*-values were calculated versus the PBS control at each time point, and are indicated above each graph. ICR mice were challenged with indicated *V. vulnificus* strains (*n* = 10, 5 × 10⁶ CFU per group), and the ventral surface temperature (**g**) and survival rates (**h**) of the infected mice are shown. h.p.i., hours post-infection. The ventral surface temperature at 14 h was analyzed, and *P*-values compared with those from PBS-treated mice. Data shown in **a**–**d** are pooled from three independent experiments (mean ± SEM). Data shown in **e** and **f** are representative of three independent experiments, each with similar results (mean ± SEM). Data shown in **g** and **h** are pooled from *n* = 5 per group from *n* = 2 independent experiments. *P*-values were calculated using one-way ANOVA with multiple comparisons (**a**, **g**), a non-parametric Mann-Whitney U-test (two-tailed) (**b**–**d**), or a two-way ANOVA with multiple comparisons (**e**, **f**) or log-rank test (**h**). ns, not significant. Source data are provided as a Source Data file.

activity of host proteins through post-translational modification (i.e., ADP-ribosylation)[62] or by depleting NAD$^+$ [63–67]. Deadly human pathogens such as *Streptococcus pyogenes* and *Mycobacterium tuberculosis* produce endogenous NADase, together with cognate immunity factors that act as an antitoxin by forming complexes with NADase to inactivate it and prevent "self-poisoning" (NAD$^+$ is also essential for metabolic and energy homeostasis in pathogens)[65,68–70]. Upon infection of macrophages, the pathogen-secreted NADase translocates to the host cytosol where it hydrolyzes NAD$^+$, thereby exerting cytotoxic effects[65,71–73].

Here, we report that, unlike NADase-secreting pathogens, MARTX toxin-secreting pathogens have evolved distinct strategies to attack their hosts. The MARTX toxin RDTND-RID effector duet takes advantage of host cell machinery to turn itself into a critical weapon that paralyzes first line immune defense mechanisms by disrupting host cell NAD$^+$ homeostasis and disabling Rac1. As suggested by its original name, DUF1 (i.e., domain of unknown function in the first position), the function of RDTND was never thought to be that of a NAD$^+$-hydrolyzing enzyme. Its structural transformation into an NADase resembling host CD38 in infected host cells, a process assisted by its closest neighboring effector (i.e., RID), is sublimely unaware of evolutionary strategy of pathogens.

Although the mechanisms underlying CD38-mediated immune responses remain poorly understood and controversial, we do know that CD38 expression is induced robustly in macrophages and monocytes in response to infection, after which it regulates inflammatory responses during the early innate immune response[74–79]. CD38, in addition to NAD$^+$-hydrolyzing enzyme, is one of the main mammalian ADP-ribosyl cyclases. CD38 generates Ca$^{2+}$-mobilizing second messengers ADPR and cADPR from NAD$^+$, and nicotinic acid adenosine dinucleotide phosphate (NAADP) from NADP$^+$ [80]. A recent study shows that CD38-mediated production of cADPR and NAADP, but not that of ADPR, induces expression of tristetraprolin (TTP), which downregulates expression of CD38 expression, leading to an increase in NAD$^+$ levels (peaking 12 h after LPS treatment) and Sirtuin 1 activation in LPS-challenged macrophages[79]. The Sirtuin 1-dependently activated TTP then suppresses acute inflammatory responses and promotes autophagic bacterial clearance during the late phase of innate immunity, highlighting the importance of timely expression and activity of CD38 for resolution of inflammation and prevention of sepsis[79]. Here, we show that in mice, *V. vulnificus* secreting a MARTX toxin harboring the RDTND-RID effector duet significantly depletes NAD(P)$^+$ levels and suppresses production of pro-inflammatory cytokines (including TNF-α) by downregulating ROS generation within 2 h post-infection (Fig. 5). This promotes dissemination of the bacteria and induces typical signs and symptoms of bacterial sepsis within 7–10 h post-infection (Fig. 6b–g). Taken together, these data suggest that the marked depletion of NAD(P)$^+$ by RDTND-RID inhibits production of Ca$^{2+}$ mobilizing second messengers during the early phase of infection, thereby paralyzing early innate immune defense mechanisms that normally promote resolution of late phase inflammation and prevent onset of sepsis.

A study reports that, unlike the MARTX$_{Vc}$ RID MLD, the corresponding domain in MARTX$_{Vv}$ RID (which is 93% similar) shows only weak membrane localization[81]. Our analysis confirmed that the entire MARTX$_{Vv}$ RID, except the RID$_{CBD}$, is required for plasma membrane targeting (Fig. 4d). These features may correlate with the evolutionary conservation of *V. vulnificus* strains that secrete MARTX toxins carrying the RDTND and the RID, rather than the RDTND alone (Fig. 1b, c); this allows concurrent targeting of two important host proteins: CaM in the cytoplasm and Rac1 at the membrane (Fig. 4e). In addition to NAD$^+$ depletion, RDTND located in close proximity to the cell membrane via RID may allow efficient depletion of NADP$^+$ (the by-product of NOX2), a process required for NADPH production (the substrate of NOX2)[43], further contributing to blockade of ROS generation.

The present study revealed that *V. cholerae* strains mainly secrete RID-containing MARTX toxins, a significant portion of which are RDTND-RID-containing toxins (Fig. 1a, b). Remarkably, nearly all clinical strains of *V. cholerae* secrete RID-carrying MARTX toxins (Fig. 1c). These results suggest that MARTX$_{Vc}$ may have evolved to robustly target Rac1. Note that MARTX$_{Vc}$ RID binds to Rac1 with an affinity that is five-fold higher than that of MARTX$_{Vv}$ RID binding to Rac1[14]. In general, the MARTX toxins secreted by *V. cholerae* strains deliver RID together with ACD (rather than with RDTND) for effective actin cytoskeleton collapse via the suppression of Rac1-mediated inflammatory responses to cytoskeletal damage, which promotes a non-inflammatory diarrheal disease (i.e., cholera)[13]. Since MARTX$_{Vc}$ RID interacts with CaM, it remains to be clarified whether CaM is involved in *V. cholerae* pathogenesis.

It is evident that NAD$^+$ homeostasis is a key determinant of core kidney function[82]. Interestingly, renal failure is a common sequelae of *V. vulnificus* infection[83]; indeed, a few cases reported in different countries highlight acute kidney injury in patients with cholera[84–86]. Our study proposes that *Vibrio*-related kidney problems may be interrelated with NAD$^+$-depletion by the RDTND-RID duet.

In summary, pathogenic bacteria constantly evolve a wide range of strategies to invade and colonize their hosts. Our findings provide mechanistic insight into how pathogenic bacteria-generated effectors can act in a mutually-dependent manner, and may prove useful for the future design of tools or strategies to combat MARTX toxin-related human diseases.

## Methods

### Ethical statement

All animal experiments were carried out according to a protocol approved by the Institutional Animal Use and Care Committee of the Korea Research Institute of Bioscience and Biotechnology (KRIBB-AEC-18186). We have complied with all relevant ethical regulations for animal use.

### Bacterial strains, culture media, cell culture, and mice

The bacterial strains, culture media, cell lines, and mice used in this study are listed in Supplementary Data 1. *Escherichia coli* strains were grown at 37 °C in Luria-Bertani medium (LB) supplemented with appropriate antibiotics. As previously reported[1,87,88], *Vibrio vulnificus* MO6-24/O strains were grown at 30 °C in LB medium supplemented with 2% (w/v) sodium chloride (LBS). HEK293T, HeLa, and RAW 264.7 cells were obtained from the American Type Culture Collection. HEK293T, HeLa, RAW 264.7, and murine bone marrow-derived macrophage (BMDM) cells were cultured at 37 °C/5% CO$_2$ in Dulbecco's modified Eagle's medium (DMEM; Corning) containing 10% (v/v) fetal bovine serum (Gibco) and antibiotic-antimycotic solution (Cytiva). All mice (specific pathogen-free; 5 or 7-week-old female CrlOri:CD1 (ICR) mice) used in this study were purchased from Orientbio and acclimatized in the institutional facility for at least 24 h before the infection experiments (12 h light/dark cycle at 22 ± 1 °C temperature with 50 ± 5% humidity).

### DNA and plasmids

Genomic DNA extracted from *V. vulnificus* MO6-24/O was used as a template for amplification of DNA fragments encoding MARTX toxin proteins. Amplified DNA fragments corresponding to the DUF1-RID effector module [referred to hereafter as RDTND-RID (residues 1959–2901] and its C-terminally-deleted proteins [RDTND-RID$_{ND/MLD}$ (residues 1959–2468); RDTND-RID$_{ND}$ (residues 1959–2374); and RDTND (residues 1959–2252)] were subcloned individually into the pProEX HTa vector expressing a 6×His-TEV cleavage site at the N-terminus (Invitrogen). The primers used are listed in Supplementary Data 2. If necessary, the DNA fragments mentioned above were subcloned individually into the pHis-Parallel1 vector[89] to express

non-tagged recombinant proteins. Human calmodulin2 (CaM) DNA was purchased from the Korea Human Gene Bank and subcloned into the pHisGST-Parallel1 vector[89] to express the protein with an N-terminal 6×His/GST tag. For expression in mammalian cells, DNA fragments encoding RDTND-RID variants [RDTND-RID; RDNTD; RID (residues 2263–2901); RDTND-RID$_{C/A}$ (mutation of C2838 to alanine); RDTND$_{E/Q}$-RID$_{C/A}$ (mutations of E2186 and C2838 to glutamine and alanine, respectively); and RDTND$_{E/Q}$-RID$_{CBD/MLD\_C}$ (residues 1959–2581); RDTND$_{E/Q}$-RID$_{CBD/MLD}$ (residues 1959–2468); RID$_{C/A}$] were subcloned individually into the pAcGFP_N1 vector (Clontech), which generates proteins carrying GFP at the C-terminus. To identify interacting partners of RDTND-RID or RID in mammalian cells, DNA encoding RDTND$_{E/Q}$-RID$_{C/A}$ or RID$_{C/A}$ was subcloned individually into pEXPR-IBA 103 (IBA Lifesciences) to express proteins carrying a strep tag at the C-terminus. DNA encoding human Ras-related C3 botulinum toxin substrate 1 (Rac1) was purchased from the Korea Human Gene Bank. DNA fragments encoding Rac1 (residues 1–192) or C-terminally-deleted Rac1 (residues 1–177) were subcloned individually into the pProEX HTa vector to express proteins carrying a 6×His-TEV cleavage site at the N-terminus. Plasmids expressing RDTND-RID mutants and Rac1 mutants (Rac1$_{Q61L}$, and Rac1$_{T17N}$) were generated by site-directed mutagenesis using the primers described in Supplementary Data 2, *Pfu* DNA polymerase (NanoHelix), and the DpnI restriction enzyme (NEB).

## Sequence analysis of MARTX toxins

MARTX toxins containing the RDTND-RID module and its variants were subjected to sequence analysis using the amino acid sequence of RDTND-RID (residues 1959–2901) of *V. vulnificus* MO6-24/O strain as a template. The full sequences of 1858 MARTX toxins were downloaded from the NCBI database and compared with that of RDTND-RID and its variants using NCBI Blast+ (version 2.14.0 + ), with the BLASTp option. Each sequence that exhibited a similarity of at least 60% (and a coverage of at least 90%) with the RDTND-RID module sequence was deemed identical. To detect the presence of a MARTX toxin containing the RDTND-RID module in clinical isolates of *V. cholerae* and *V. vulnificus*, a total of 5979 and 1373 genomes, respectively, that had been assembled at least to the scaffold-level or higher were downloaded from the NCBI Genome database. Only genomes originating from *Homo sapiens* were downloaded from the NCBI BioSample database (which lists the origin of strains) for use in this study. The representative RDTND-RID module sequence from *V. vulnificus* MO6-24/O was aligned to each genome using NCBI Blast+ (version 2.14.0 + ), with the BLASTn option. Genome sequences exhibiting at least 90% coverage of the RDTND-RID module or RID were considered as matches, and subsequently classified.

## Expression and purification of recombinant proteins

*E. coli* BL21-CodonPlus (DE3) RIPL strain (Agilent) was transformed with plasmids containing the 6×His-tagged RDTND-RID module or its variants [RDTND-RID$_{C/A}$; RDTND$_{E/Q}$-RID$_{C/A}$; RDTND$_{4mt}$-RID$_{C/A}$ (mutations: W2183, R2195, E2285, and R2328 to leucine for RDTND$_{4mt}$); RDTND-RID$_{CBD}$; and RDTND]. *E. coli* strains were grown at 37 °C in LB broth containing 100 μg mL$^{-1}$ ampicillin until the optical density at 600 nm (OD$_{600}$) reached 0.5–0.6. Then, expression of recombinant proteins was induced at 18 °C by addition of 0.5 mM isopropyl β-D-thiogalactopyranoside (IPTG; LPS Solution) for 18 h. The cells were harvested by centrifugation at 4000 × *g* for 10 min, resuspended in buffer A [50 mM Tris-HCl (pH 7.5), 300 mM NaCl, and 5% (v/v) glycerol] supplemented with 10 mM imidazole, and then lysed using a high pressure homogenizer (Nano DeBee). After centrifugation at 14,000 × *g* for 1 h at 4 °C, the supernatants were subjected to immobilized metal affinity chromatography (IMAC) using a nickel-nitrilotriacetic acid (Ni-NTA) resin (Qiagen) pre-equilibrated with buffer A. The column was washed with buffer A supplemented with 25 mM imidazole, and 6×His-

tagged recombinant proteins bound to the resin were eluted with buffer A supplemented with 250 mM imidazole. When necessary, the N-terminal 6×His-tag was removed from proteins by treatment with TEV protease (Sigma Aldrich) for 18 h at 4 °C during dialysis into buffer B [50 mM Tris-HCl (pH 7.5), 150 mM NaCl, and 5% (v/v) glycerol]. Proteins not digested by TEV proteases were removed by a second affinity chromatography step using Ni-NTA resin, and eluted proteins were further purified by size-exclusion chromatography using a HiLoad 16/60 Superdex™ 200 prep grade column (GE Healthcare) pre-equilibrated with buffer B. To purify 6×His/GST-fused CaM in the presence of CaCl$_2$ (6×His/GST-CaM/Ca$^{2+}$), harvested *E. coli* cells expressing 6×His/GST-fused CaM were lysed in buffer A supplemented with 5 mM CaCl$_2$ and 10 mM imidazole, and further purified using the procedures described above. To purify 6×His/GST-fused CaM in the absence of CaCl$_2$ (6×His/GST-CaM), harvested *E. coli* cells were lysed in buffer C [50 mM Tris-HCl (pH 7.5), 150 mM NaCl, 10 mM EDTA, and 10 mM EGTA] using a high pressure homogenizer (Nano DeBee). After centrifugation at 14,000 × *g* at 4 °C for 1 h, the supernatant was subjected to immobilized affinity chromatography using Glutathione Sepharose™ 4 Fast Flow resin (Cytiva) pre-equilibrated with buffer C. The column was washed with buffer C, and 6×His/GST-fused recombinant proteins bound to the resin were eluted with buffer C supplemented with 30 mM reduced glutathione. The 6×His/GST tags were cleaved from the recombinant proteins by TEV protease, and non-digested proteins and 6×His/GST were removed by a second affinity chromatography step using Ni-NTA resins. The eluted proteins were further purified by size-exclusion chromatography using a HiLoad 16/60 Superdex™ 75 prep grade column (GE Healthcare) pre-equilibrated with buffer B. To prepare Ca$^{2+}$-bound CaM in complex with RDTND-RID$_{C/A}$ or RDTND-RID$_{CBD}$, harvested cells expressing 6×His/GST-CaM and harvested cells expressing RDTND-RID variants without a tag were mixed together and lyzed in buffer A supplemented with 5 mM CaCl$_2$ and 10 mM imidazole. Proteins were purified as described for 6×His/GST-CaM/Ca$^{2+}$. To purify CaM complexed with RDTND-RID$_{C/A}$ or RDTND-RID$_{CBD}$, harvested cells expressing 6×His/GST-CaM or RDTND-RID variants without a tag were mixed together, lyzed in buffer C, and purified as described for 6×His/GST-CaM. To prepare selenomethionine (SeMet)-labeled proteins (RDTND-RID$_{CBD}$, RDTND-RID$_{CBD}$/CaM, and RDTND-RID$_{CBD}$/CaM/Ca$^{2+}$), *E. coli* B834 (DE3) strain (Novagen) was transformed with the corresponding expression plasmids, and grown in minimal M9 medium supplemented with 50 mg mL$^{-1}$ of SeMet and 100 μg mL$^{-1}$ of ampicillin until OD$_{600}$ = 0.8. Expression of SeMet-labeled proteins was induced at 25 °C by addition of 0.5 mM IPTG until OD$_{600}$ = 1.5 (at least). SeMet-labeled proteins were purified as described above. All purified proteins were concentrated to 15–20 mg mL$^{-1}$ using an Amicon Ultra-15 30 K device (EMD Millipore) prior to crystallization and kept at −80 °C until use. To purify Rac1 proteins, harvested *E. coli* cells expressing N-terminal 6×His-tagged Rac1 proteins (Q61L mutant for GTP-bound form and T17N mutant for GDP-bound form) were lyzed in buffer A supplemented with 1 mM 1,4-dithiothreitol (DTT; LPS Solution) and 10 mM imidazole, and purified as described for the 6×His-tagged RDTND-RID module. To purify the RDTND-RID$_{C/A}$ complexed with Ca$^{2+}$-bound CaM and Rac1$_{Q61L}$, *E. coli* cells expressing RDTND-RID$_{C/A}$ and 6×His/GST-CaM were mixed together and lyzed in buffer A supplemented with 5 mM CaCl$_2$ and 10 mM imidazole. Meanwhile, *E. coli* cells expressing 6×His-tagged Rac1$_{Q61L}$ (residues 1–177) were lyzed in buffer A supplemented with 1 mM DTT and 10 mM imidazole. Then, each sample was purified as described above. All purified proteins were pooled in a beaker and incubated with gentle stirring for 3 h at 4 °C. After incubation, the RDTND-RID$_{C/A}$/CaM/Rac1$_{Q61L}$ complex was further purified using a Superdex™ 200 10/300 GL column (GE Healthcare) pre-equilibrated with buffer D [50 mM HEPES-NaOH (pH 7.5), 150 mM NaCl, and 1% (v/v) glycerol] prior to cryo-EM.

### In vitro pull-down assay

To investigate the interaction between RDTND-RID variants and CaM, 0.5 mg of each purified protein (RDTND-RID, RDTND-RID$_{ND/MLD}$, RDTND-RID$_{ND}$, or RDTND), and 0.25 mg of purified 6×His/GST-CaM, were used. The protein mixtures were reacted at 4 °C for 1 h in TBS buffer [50 mM Tris-HCl (pH 7.4) and 137 mM NaCl] containing 10 mM EDTA and 10 mM EGTA. Then, the mixtures were loaded onto 0.1 mL of Glutathione Sepharose™ 4 Fast Flow affinity resins. The resins were washed with a 150-fold resin volume of TBS buffer, and samples were eluted with TBS buffer supplemented with 50 mM reduced glutathione. Eluates were analyzed by SDS-PAGE and visualized using Coomassie Brilliant Blue staining. To validate the Ca$^{2+}$-dependency of the interaction between RDTND-RID and CaM, 0.5 mg of RDTND-RID module protein was incubated with 0.25 mg of 6×His/GST-CaM or 6×His/GST-CaM/Ca$^{2+}$ in TBS buffer containing 10 mM EDTA and 10 mM EGTA, or in TBS buffer containing 10 mM CaCl$_2$, respectively, at 4 °C for 1 h.

### Isothermal titration calorimetry

To measure the binding affinity of CaM or CaM/Ca$^{2+}$ for the RDTND-RID module, purified CaM, CaM/Ca$^{2+}$, or RDTND-RID$_{C/A}$ proteins were dialyzed in buffer B at 4 °C for 18 h. All purified proteins were degassed at 25 °C for 20 min using a ThermoVac (Microcal). CaM or CaM/Ca$^{2+}$ (450 µM) was loaded into the syringe and titrated against 30 µM RDTND-RID$_{C/A}$ in the reaction cell. Isothermal titration calorimetry measurements were performed at 25 °C, with an injection volume of 6 µL, 35 injections with an interval of 240 s between each, and a stirring speed of 300 rpm. Heat data derived from injection of CaM or CaM/Ca$^{2+}$ into buffer B were subtracted from those derived from injection of CaM or CaM/Ca$^{2+}$ into RDTND-RID$_{C/A}$. Raw data were processed, fitted, and plotted using the Origin software (version 7) supplied with the VP-ITC Microcalorimeter (Microcal).

### Crystallization

Purified SeMet-labeled RDTND-RID$_{CBD}$ proteins were screened initially at 20 °C prior to crystallization using the sitting-drop vapor-diffusion method conducted in 96-well MRC crystallization plates (Swissci); proteins were crystallized in a reservoir solution containing 8% (v/v) Tacsimate (pH 6.0) and 20% (w/v) polyethylene glycol (PEG) 3350 from the PEG/Ion screening kit (Hampton Research). Optimized SeMet-labeled RDTND-RID$_{CBD}$ crystals were grown at 20 °C in 2 µL droplets containing equal volumes of protein and reservoir solution [4% (v/v) Tacsimate (pH 6.0) and 22% (w/v) PEG 3350]. For data collection, SeMet-labeled RDTND-RID$_{CBD}$ crystals were cryoprotected by transferring them into reservoir solution containing an additional 10% (v/v) ethylene glycol and flash-frozen in liquid nitrogen. After crystallization of complex proteins using the sitting-drop vapor-diffusion method in 96-well MRC crystallization plates (Swissci), RDTND-RID$_{CBD}$/CaM/Ca$^{2+}$ and SeMet-labeled RDTND-RID$_{CBD}$/CaM/Ca$^{2+}$ microcrystals were generated at 20 °C after 3 days in reservoir solution containing 20% (w/v) PEG 3350 and 0.2 M magnesium acetate from the MCSG1 screening kit (Anatrace). Optimized RDTND-RID$_{CBD}$/CaM/Ca$^{2+}$ and SeMet-labeled RDTND-RID$_{CBD}$/CaM/Ca$^{2+}$ crystals were grown at 20 °C in 2 µL droplets containing equal volumes of protein and reservoir solution [17% (w/v) PEG 3350 and 0.18 M magnesium acetate, or 20% (w/v) PEG 3350 and 0.17 M magnesium acetate, respectively]. For data collection, RDTND-RID$_{CBD}$/CaM/Ca$^{2+}$ and SeMet-labeled RDTND-RID$_{CBD}$/CaM/Ca$^{2+}$ crystals were cryoprotected by transferring them into a reservoir solution containing an additional 10% (v/v) ethylene glycol and flash-frozen in liquid nitrogen. SeMet-labeled RDTND-RID$_{CBD}$/CaM crystals were produced at 20 °C after 1 day in reservoir solution containing 25% (w/v) polyethylene glycol (PEG) 3350, 0.1 M HEPES-NaOH (pH 7.5), and 0.2 M MgCl$_2$ from the MCSG1 screening kit (Anatrace). Optimized RDTND-RID$_{CBD}$/CaM crystals were generated at 20 °C in 2 µL droplets containing equal volumes of protein and reservoir solution [20% (w/v) PEG

3350, 0.1 M HEPES-NaOH (pH 7.5), 0.28 M MgCl$_2$, and 6 mM MnCl$_2$]. For data collection, crystals were flash-frozen in liquid nitrogen. To determine the structure of the NAD$^+$-bound RDTND$_{E2186Q}$-RID$_{CBD}$/CaM/Ca$^{2+}$ complexes, proteins were screened initially for crystallization using the sitting-drop vapor-diffusion method in 96-well MRC crystallization plates. RDTND$_{E2186Q}$-RID$_{CBD}$/CaM/Ca$^{2+}$ microcrystals were produced at 20 °C after 2 days in reservoir solution [0.1 M Tris-HCl (pH 8.5), 30% (w/v) PEG 4000, and 0.2 M sodium acetate] from the SG1™ Screen screening kit (Molecular Dimensions). Optimized RDTND$_{E2186Q}$-RID$_{CBD}$/CaM/Ca$^{2+}$ crystals were grown at 20 °C in 2 µL droplets containing equal volumes of protein and reservoir solution [0.1 M Tris-HCl (pH 8.5), 32.5% (w/v) PEG 4000, and 0.15 M sodium acetate]. RDTND$_{E2186Q}$-RID$_{CBD}$/CaM/Ca$^{2+}$ was soaked for 2 h in a soaking solution containing an additional 10 mM NAD$^+$ (Sigma Aldrich). For data collection, NAD$^+$-bound RDTND$_{E2186Q}$-RID$_{CBD}$/CaM/Ca$^{2+}$ crystals were mounted directly on the goniometer.

### Collection of X-ray diffraction data and structure determination

SeMet-labeled RDTND-RID$_{CBD}$/CaM/Ca$^{2+}$ crystals were diffracted at a resolution of 3.05 Å, and single wavelength anomalous dispersion (SAD) data were collected at beamline 5 C at the Pohang Acceleratory Laboratory (PAL). All data were indexed, processed, and scaled using the HKL2000 software package[90]. The initial phases for SeMet-labeled RDTND-RID$_{CBD}$/CaM/Ca$^{2+}$ were obtained by the molecular replacement (MR)-SAD method using the Phenix program[91], with the structure of human CaM (PDB ID 1CLL) as the MR template. Crystals of SeMet-labeled RDTND-RID$_{CBD}$ were diffracted at a resolution of 3.38 Å, and SAD data were collected at beamline 5 C at PAL. The initial phases of SeMet-labeled RDTND-RID$_{CBD}$ were determined by the MR-SAD method using Phenix[91], with the structure of RDTND-RID$_{CBD}$ within the SeMet-labeled RDTND-RID$_{CBD}$/CaM/Ca$^{2+}$ complex used as the MR template. Crystals of the NAD$^+$-bound RDTND$_{E2186Q}$-RID$_{CBD}$/CaM/Ca$^{2+}$ were diffracted at beamline 5 C at PAL at a resolution of 2.35 Å. The structure of NAD$^+$-bound RDTND$_{E2186Q}$-RID$_{CBD}$/CaM/Ca$^{2+}$ was determined by MR using Molrep in the CCP4i suite using the RDTND-RID$_{CBD}$/CaM/Ca$^{2+}$ structure as the MR template. Crystals of SeMet-labeled RDTND-RID$_{CBD}$/CaM were diffracted at beamline 11 C of PAL at a resolution of 2.82 Å, and SAD data were collected at beamline 11 C at PAL. All diffraction data were indexed, processed, and scaled using the HKL2000 software package and the XDS program package. The initial phases of SeMet-labeled RDTND-RID$_{CBD}$/CaM were obtained by the MR method using Molrep in the CCP4i suite, with the SeMet-labeled RDTND-RID$_{CBD}$/CaM/Ca$^{2+}$ used as the MR template. Model building was carried out using Coot[92], and refinement was performed by Refmac5[93]. Ramachandran statistics were analyzed by MolProbity[94]. All data collection and refinement statistics are summarized in Supplementary Table 1. Superimposition of structures was carried out using PyMOL align (Schrödinger) and Coot SSM superpose. The Dali server was used to identify proteins with structures similar to the RDTND-RID$_{CBD}$ of RDTND-RID$_{CBD}$/CaM/Ca$^{2+}$. All molecular graphics were generated using PyMOL version 1.5.0.4 (Schrödinger) and ChimeraX version 1.5[95].

### Preparation of samples for cryo-EM and data acquisition

For the cryo-EM studies of native samples, the purified RDTND-RID$_{C/A}$/CaM/Rac1$_{Q61L}$ complex (0.75 mg mL$^{-1}$) was applied to a glow-discharged holey-carbon copper grid (Quantifoil, R1.2/1.3, 200-mesh; EMS). The grids were blotted for 3.0 s and plunge-frozen in ethane cooled with liquid nitrogen using Vitrobot Mark IV (FEI), with the environmental chamber set at a humidity of 100% and a temperature of 4 °C. Automatic cryo-EM data were collected on a Glacios (FEI) at an acceleration voltage of 200 kV using EPU software (FEI), and images were recorded using a Falcon IV direct electron detector (FEI). A total of 12,472 micrographs were recorded at a defocus range of −1.4 to −2.2 µm, with a nominal magnification of 120,000, resulting in a physical

pixel size of 0.87 Å. Each image was dose-fractionated into 50 movie frames, with a total exposure time of 6.98 s, resulting in a total dose of 55 electrons per A².

For the cryo-EM studies of cross-linked samples, the purified RDTND-RID$_{C/A}$/CaM/Rac1$_{Q61L}$ complex proteins were incubated for 2.5 h at 4 °C in buffer D containing 0.5% (v/v) glutaraldehyde. Then, the cross-linked proteins were loaded to a 5–20% glycerol gradient and ultracentrifuged at 160,000 × *g* at 4 °C for 18 h using a swinging bucket rotor (Eppendorf Himac Technologies). After ultracentrifugation, the samples were fractionated with a Piston Gradient Fractionator™ (Biocomp Instruments), and the fraction containing homogenous cross-linked proteins was further purified by size-exclusion chromatography using a Superdex™ 200 10/300 GL column (GE Healthcare). The procedures for purification are shown in Supplementary Fig. 8. The prepared sample (0.75 mg mL$^{-1}$) was applied to a glow-discharged holey-carbon copper grid (Quantifoil, R1.2/1.3, 300-mesh; EMS). The grids were blotted for 3.0 s and plunge-frozen in ethane cooled with liquid nitrogen using Vitrobot Mark IV (FEI), with the environmental chamber set at a humidity of 100% and a temperature of 4 °C. Automatic cryo-EM data were collected on a Titan Krios G4 (FEI) at an acceleration voltage of 300 kV using EPU software (FEI), and images were recorded using a BioQuantum K3 direct electron detector (Gatan). A total of 11,236 micrographs were recorded at a defocus range of −1.4 to −2.2 μm, with a nominal magnification of 130,000, resulting in a physical pixel size of 0.66 Å. Each image was dose-fractionated into 50 movie frames, with a total exposure time of 2.8 s, resulting in a total dose of 50 electrons per A² (Supplementary Table 2).

## Processing of cryo-EM data

The procedures used for data processing of cross-linked samples are shown in Supplementary Fig. 9. All data processing was done in cryoSPARC version 4.1.2[96]. Motion correction and contrast transfer function estimation were done in cryoSPARC version 4.1.2. The initial 4,482,761 particles were picked automatically from 11,236 micrographs using Blob picker. Next, 179,094 particles were selected by Extract and repetitive 2D Classification, and classified to generate 2D templates. For further automated particle picking from the 179,094 particles, TOPAZ Train & Extract was used, with a box size of 300 pixels, and 3,357,262 particles were picked from 11,236 micrographs. The auto-picked particles were 2D classified over multiple rounds, and 437,183 particles were separated from all particles in bad 2D classes. The particles were re-extracted with a box size of 500 pixels and 2D classified again through multiple rounds. The selected 247,635 particles were then used for ab initio 3D reconstruction and subjected to heterogeneous 3D refinement. There was no significant difference between the four 3D reconstruction maps obtained from the heterogeneous 3D refinement. All 247,635 particles was subjected to several rounds of Non-uniform 3D Refinement, yielding a final reconstruction of the RDTND-RID$_{C/A}$/CaM/Rac1$_{Q61L}$ complex at 4.32 Å. The resolution was measured based on the gold-standard Fourier shell correlation cut-off criterion of 0.143. The procedures used for data processing of native samples are shown in Supplementary Fig. 10. All data processing was done in cryoSPARC version 4.3.1[96]. Motion correction and contrast transfer function estimation were done in cryoSPARC version 4.3.1. The initial 1,869,419 particles were picked automatically from 11,558 micrographs using Blob picker. Next, 47,009 particles were selected by Extract and repetitive 2D Classification, and classified to generate 2D templates. For further automated particle picking from the 47,009 particles, TOPAZ Train & Extract was used with a box size of 300 pixels, and 1,437,704 particles were picked from 11,558 micrographs. The auto-picked particles were 2D classified over multiple rounds, and 181,398 particles were separated from all particles in bad 2D classes. The selected 2D classified particles showed low resolution due to high flexible movement.

## Fitting of models into cryo-EM map

The L-shaped overall feature of the RDTND-RID$_{C/A}$/CaM/Rac1$_{Q61L}$ complex was observed in a cryo-EM map contoured at 4.5 σ. Individual X-ray structures of RDTND-RID$_{CBD}$ complexed with CaM (this study), RID$_{MLD\_C/CD}$ (PDB ID: 5XN7), and active Rac1 (PDB ID: 3TH5) were well docked in the cryo-EM map using ChimeraX version 1.5[95]. In detail, the structure of RDTND-RID$_{CBD}$ complexed with CaM was docked in the cryo-EM density map contoured at ~8–12 σ. The structures of RID$_{MLD\_C/CD}$ and the active mimic Rac1$_{Q61L}$ (residues, 1–177) were fitted in the cryo-EM density map contoured at ~4.5–6 σ (Supplementary Fig. 11). The C$_\alpha$-backbones were fitted in the cryo-EM map; however, the RID flexible region (residues, 2702–2764) and the side chains of the amino acids could not be modeled due to a weak cryo-EM map. The structural regions of RID$_{MLD\_C/CD}$ and Rac1$_{Q61L}$ were further refined using GalaxyRefineComplex on the GalaxyWEB server[97], based on repeated structure perturbations and subsequent overall structural relaxation by short molecular dynamic simulation. The overall structure of RDTND-RID$_{C/A}$/CaM/Rac1$_{Q61L}$ was finally fitted into the cryo-EM map by WinCoot[92] and phenix.cryo_fit, using Phenix version 1.20.1–14487[91,98]. The structural model of the complex fitted into the cryo-EM map is provided in Supplementary Data 3.

## Confocal microscopy

To assess the cytotoxicity of RDTND-RID variants, plasmids expressing GFP or C-terminally GFP-fused RDTND-RID, RDTND-RID$_{C/A}$, RDTND, RID, or RID$_{C/A}$ were transiently transfected into $1 \times 10^5$ of HEK293T cells in an 8-well chamber slide (Ibidi) using a Lipofectamine™ 3000 transfection reagent (Thermo Fisher Scientific). After 18 h, cells were washed twice in Dulbecco's phosphate-buffered saline (DPBS; Cytiva) and fixed for 20 min at room temperature using 4% (v/v) formaldehyde (Thermo Fisher Scientific) in DPBS. Next, 50% methanol in DPBS was added, followed by permeabilization in 100% methanol at −20 °C for 20 min. To analyze the subcellular localization of the RDTND-RID variants, $2 \times 10^4$ HeLa cells in an 8-well chamber slide (Ibidi) were transfected with plasmids expressing GFP, or C-terminally GFP-fused RDTND$_{E/Q}$-RID$_{C/A}$, RDTND$_{E/Q}$-RID$_{MLD\_C}$, RDTND$_{E/Q}$-RID$_{MLD}$, or RDTND$_{E/Q}$, using X-tremeGENE™ HP DNA transfection reagent (Roche). After 24 h, cells were washed twice with DPBS. Prepared cells were stained with corresponding reagents (Hoechst 33342 for nuclei, Thermo Fisher Scientific; Rhodamine phalloidin for F-actin, Thermo Fisher Scientific; CF® 594-conjugated wheat germ agglutinin (WGA) for the plasma membrane; Biotium). Images were obtained under a C2plus laser scanning confocal microscope (Nikon) and analyzed using NIS-Elements software (Nikon).

## Nuclear magnetic resonance (NMR) analysis

Purified RDTND-RID complexed with CaM (66 nM) was incubated with 1 mM NAD$^+$ in NMR reaction buffer [20 mM NaH$_2$PO$_4$/Na$_2$HPO$_4$ (pH 7.4), and 80 mM NaCl] in a final volume of 500 μL at room temperature for 30 min. The $^1$H NMR spectra of the reaction products were acquired on an Avance 700 MHz spectrometer (Bruker) equipped with a cryogenic triple resonance probe, operating at a proton frequency of 700 MHz at 298 K. The acquisition was carried out using Topspin software version 3.5 (Bruker) and a zgesgp pulse program (Bruker). Each spectrum for each time point was obtained with 4 scans. The spectral width was 9803.922 Hz, and the acquisition time was 835.584 ms. The spectra were processed with a window function of a 90°-shifted sine bell using Topspin software version 3.5.

## Liquid chromatography-tandem mass spectrometry (LC-MS/MS)

To identify RID-interacting proteins in mammalian cells, individual HEK293T cells were transfected with either empty vector (pEX-PR_IBA103) or plasmid expressing C-terminally strep-tagged RID$_{C/A}$, and then incubated for a further 48 h to allow protein expression. Then, the cells were harvested and lysed with RIPA lysis buffer [50 mM

Tris-HCl (pH 7.5), 100 mM NaCl, 1% (v/v) NP-40, 1 mM EDTA, 0.25% (w/v) Na-deoxycholate, and protease and phosphatase inhibitor cocktail (GenDepot)]. Cell lysates were centrifuged at 16,000 × $g$ at 4 °C for 20 min, and the supernatants were incubated at 4 °C for 3 h with 50 µL of Strep-Tactin® Sepharose® resin (IBA Lifesciences), with gentle agitation. The supernatant was removed from the resin, and the resin was washed with a 200-fold resin volume (10 mL) of Strep wash buffer [100 mM HEPES-NaOH (pH 8.0), 150 mM NaCl, and 1 mM EDTA]. The proteins bound to the resin were eluted with Strep elution buffer [100 mM Tris-HCl (pH 8.0), 150 mM NaCl, 1 mM EDTA, and 2.5 mM desthiobiotin]. Samples ($n$ = 1 for empty vector and RID$_{C/A}$-Strep) were digested using Straps micro (PROTIFI) following the manufacturer's protocol version 4.7, which includes the use of tris(2-carboxyethyl) phosphine (TCEP) and iodoacetamide (IAA) for reduction and alkylation. One µg of trypsin/lysC (Pierce) was added to each sample and incubated overnight at 37 °C. The eluted samples were dried with a speed-vac. The peptides were then resuspended in a 2% acetonitrile/ 0.1% formic acid solution and injected into a U3000 RSLCnano/Orbitrap Exploris 240 system (Thermo Fisher Scientific). The trap column used was an Acclaim™ PepMap™ 100 C18 (75 µm × 2 cm, 3 µm; Thermo Fisher Scientific), and the analytical column was a Double nanoViper™ PepMap™ Neo (75 µm × 50 cm, 2 µm; Thermo Fisher Scientific). Both columns were maintained at 60 °C. The mobile phases were 0.1% formic acid in HPLC-grade water (buffer X) and 0.1% formic acid in acetonitrile (buffer Y). The following gradient was used at a flow rate of 300 nL min$^{-1}$: 2% to 22% buffer Y over 100 min, then 22% to 35% buffer Y over 20 min. The survey scan settings were as follows: Resolution = 120,000, Max IT = 50 ms, AGC target = 300%, and mass range = 350–1200 Th. The selected precursor ions were fragmented by HCD and analyzed by the Orbitrap. Additional parameters for the MS/MS scan were as follows: Top15 data-dependent acquisition, resolution = 30,000, Max IT = 200 ms, AGC target = standard, intensity threshold = 1 × 10$^5$, normalized collision energy = 30%, isolation width = 2, dynamic exclusion after 1 occurrence, exclusion duration = 20 sec, and mass tolerance window = ± 10 ppm. LC-MS/MS raw data were analyzed using MaxQuant version 2.4.0.0[99] and Perseus version 2.0.6.0[100]. The MaxQuant parameters included using the Uniprot Homo sapiens database, setting the enzyme to Trypsin/P, and allowing for variable modifications such as Oxidation (M) and Acetylation (protein N-term). Carbamidomethylation (C) was set as a fixed modification, and the match between runs feature was enabled. After reading the data into Perseus, protein groups were filtered out based on the following criteria: reverse sequences, proteins identified only by site, contaminants, and those with fewer than two razor + unique peptides. The LC-MS/MS results for the RID$_{C/A}$-interacting candidates are listed in Supplementary Data 4. To analyze the product of NAD(P)$^+$ hydrolyzed by RDTND-RID module, 1 mM NAD(P)$^+$ was incubated with 1 µM RDTND-RID$_{C/A}$/ CaM protein in TBS buffer at 37 °C for 1 h. The reaction mixture was loaded onto an Amicon Ultra-0.5 3 K device (EMD Millipore) to remove proteins and then centrifuged at 14,000 × g at 4 °C for 20 min. The flow-through was diluted (1:5 ratio) with distilled water and analyzed by an Acquity UPLC/Synapt HDMS (Waters) system fitted with an Acquity UPLC BEH C18 column (2.1 mm × 50 mm, 1.7 µm; Waters). The mobile phases were buffer X and buffer Y, with the column temperature set to 35 °C. The gradient elution was performed at a flow rate of 0.2 mL min$^{-1}$ with the following program: 0% buffer Y for 5 min, 0% to 10% buffer Y over 1 min, and 10% to 40% buffer Y over 1 min. The survey scan settings included a mass range of 300–2000 Th, positive ion mode, and a scan time of 1 sec.

## High-performance liquid chromatography (HPLC)

To analyze the by-products of NAD$^+$ hydrolysis by CD38 or RDTND-RID, HPLC experiments were performed using a modified method from previous study[31]. One mM of NAD$^+$ was incubated at 37 °C for 30 min with CD38 (10 nM; Sigma Aldrich) or RDTND-RID/CaM complex (80 nM) in TBS buffer. The reaction was placed in an Amicon Ultra-0.5 3 K device and centrifuged at 14,000 × $g$ at 4 °C for 20 min. Then, 40 µL of the flow-through fraction was injected into a SUPELCOSIL™ LC-18 HPLC column (25 cm × 4.6 mm, 5 µm; Supelco) in a HPLC system equipped with a diode array detector. The elution conditions were as follows: starting with 9 min at 100% buffer 1 (0.1 M potassium phosphate buffer, pH 6.0), followed by a gradual increase over 6 min to 12% buffer 2 (buffer 1 with 10% methanol). Then, the gradient increased to 45% buffer 2 over 2.5 min, and further to 100% buffer 2 over the next 2.5 min. The column was then held at 100% buffer 2 for 5 min. The flow rate was maintained at 1.0 mL min$^{-1}$, and the assay was performed at room temperature. The absorbances of NAD$^+$, ADPR, and cADPR were evaluated at a wavelength of 254 nm.

## Engineering of *V. vulnificus*

To clarify the function of the RDTND-RID module within the MARTX toxins, four exotoxin gene-deleted (*vvhA* encoding hemolysin, *vvpE* encoding elastase, *plpA* encoding phospholipase A2, and *vvpM* encoding secretory protease) strains were used as the parental strains from which other *V. vulnificus* mutant strains were engineered as described previously[1]. The plasmid pDS_EF-MARTX, which contains an EF rtxA1 gene (EF-rtxA1)[1], was used to engineer plasmids pDS_RDTND-RID, pDS_RDTND-RID$_{C/A}$, pDS_RDTND$_{E/Q}$-RID$_{C/A}$, and pDS_RDTND$_{4mt}$-RID$_{C/A}$. Amplified DNA fragments of RDTND-RID, RDTND-RID$_{C/A}$, RDTND$_{E/Q}$-RID$_{C/A}$, or RDTND$_{4mt}$-RID$_{C/A}$ were subcloned into the XhoI restriction enzyme site of pDS_EF-MARTX. Consequently, plasmids pDS_RDTND-RID, pDS_RDTND-RID$_{C/A}$, pDS_RDTND$_{E/Q}$-RID$_{C/A}$, and pDS_RDTND$_{4mt}$-RID$_{C/A}$ were constructed. These constructs were delivered to the parental strain to exchange the rtxA1 locus for RDTND-RID, RDTND-RID$_{C/A}$, RDTND$_{E/Q}$-RID$_{C/A}$, or RDTND$_{4mt}$-RID$_{C/A}$ by in-frame double homologous recombination, as previously described[1], thereby generating the EF::RDTND-RID, EF::RDTND-RID$_{C/A}$, EF::RDTND$_{E/Q}$-RID$_{C/A}$, and EF::RDTND$_{4mt}$-RID$_{C/A}$ strains, respectively (Fig. 5a).

## Isolation of bone marrow-derived macrophages (BMDMs)

BMDMs were isolated from the thigh bones of 7-week-old female ICR mice and filtered through a cell strainer (40 µm; Falcon). The filtrate was treated for 10 min with ACK lysis buffer (Gibco) to remove red blood cells and centrifuged at 300 × $g$ for 5 min at room temperature. Then, the isolated BMDMs were washed with PBS, cultured in DMEM containing 10% (v/v) FBS and 50 ng mL$^{-1}$ GM-CSF (R&D Systems), and allowed to differentiate for 6 days in fresh complete medium, which was replaced every other day. The differentiated BMDMs were plated at 1 × 10$^6$ /well in 6-well plates and used for bacterial infection experiments.

## *V. vulnificus* infection

*V. vulnificus* variants were grown in LBS until OD$_{600}$ = 0.5. Bacteria were harvested by centrifugation at 2000 × $g$ at room temperature. The cell pellet was washed and diluted in PBS to 1 × 10$^8$ colony-forming units (CFU) mL$^{-1}$. Prior to infection of BMDMs, the culture medium was replaced with DMEM without FBS and antibiotic-antimycotic solution. Then, the prepared bacterial resuspension was added to the BMDMs in 6-well plates at the multiplicity of infection (MOI) of 5 and centrifuged in a swing rotor at 125 × $g$ for 3 min at room temperature. BMDMs treated with *V. vulnificus* were incubated for 2 h (or other designated time) at 37 °C/5% CO$_2$. For the mouse survival tests, 50 µL of bacterial suspension was subcutaneously injected into the region beneath the skin over the backs of the necks of 5-week-old female ICR mice (5 × 10$^6$ CFU; $n$ = 10 per group), which were under slight isoflurane anesthesia. Infected mice were observed every 2 h until 48 h post-infection. If necessary, ventral surface temperature was measured every 2 h using a non-contact infrared thermometer (Daihan Scientific)[101].

## Isolation of peripheral blood mononuclear cells (PBMCs)

To isolate PBMCs from murine blood, 5-week-old female ICR mice infected by *V. vulnificus* ($n = 8$ for PBS; $n = 12$ for EF::RDTND-RID, EF::RDTND-RID$_{C/A}$, EF::RDTND$_{E/Q}$-RID$_{C/A}$, and EF::RDTND$_{4mt}$-RID$_{C/A}$) were sacrificed at 7 h post-infection, and blood was collected from the infraorbital vein using heparin-coated capillary tubes (Kimble). Blood samples were diluted with PBS supplemented with 2% (v/v) FBS (up to 2 mL) and transferred to SepMate™-15 in vitro diagnostic tubes (StemCell Technologies) filled with 4.5 mL Ficoll-Paque™ PREMIUM 1.084 (Cytiva) for isolation of PBMCs. Tubes were centrifuged at $1200 \times g$ for 10 min and plasma containing PBMCs was transferred to a new 15 mL tube. The plasma was then washed by adding 5 mL of $1 \times$ PBS supplemented with 2% (v/v) FBS. Tubes were further centrifuged at $300 \times g$ for 8 min, and the pellet cell was used for subsequent experiments.

## NAD(P)$^+$-hydrolyzing assay

To measure the NAD(P)$^+$-hydrolyzing activity of RDTND-RID proteins, 1 µM of NAD$^+$ or NADP$^+$ (Sigma Aldrich) was incubated with 10 nM of protein in TBS buffer for 30 min at 37 °C. After completing the reaction, the remaining NAD$^+$ was measured using EnzyChrom™ NAD$^+$/NADH assay kit (Bioassay Systems) or the EnzyChrom™ NADP$^+$/NADPH assay kit (Bioassay Systems). To measure the cellular NAD$^+$ or NADP$^+$ level in BMDMs, *V. vulnificus* was added to $1 \times 10^6$ of BMDMs in a 6-well plate at an MOI of 5, and then incubated at 37 °C for 2 h. The medium was exchanged for DMEM supplemented with 100 µg mL$^{-1}$ of gentamycin to eliminate *V. vulnificus*. After further incubation for 4 h, the medium was removed and the BMDMs were washed with DPBS. Then, the intracellular NAD$^+$ or NADP$^+$ level was measured using an NAD$^+$/NADH assay kit or NADP$^+$/NADPH assay kit, respectively. Briefly, to measure intracellular NAD$^+$ or NADP$^+$ levels, BMDMs infected with *V. vulnificus* were homogenized in 100 µL of NAD(P)$^+$ extraction buffer and incubated at 60 °C for 5 min. Then, the samples were neutralized by adding 20 µL of Assay buffer and 100 µL of NAD(P)H extraction buffer, and centrifuged at $16,000 \times g$ for 5 min to remove cell debris. After centrifugation, the supernatant was used for the NAD$^+$/NADH or NADP$^+$/NADPH assay. To measure intracellular NAD$^+$ levels in PBMCs, the number of isolated PBMCs from *V. vulnificus*-infected mice as described above was counted using a Luna-II™ automated cell counter (Logos Biosystems) after trypan blue staining. Then, the intracellular NAD$^+$ level in PBMCs was measured using an EnzyChrom™ NAD$^+$/NADH assay kit and normalized to the number of PBMCs.

## Enzyme-linked immunosorbent assay (ELISA)

To analyze pro-inflammatory cytokines, supernatants from in vitro-cultured BMDMs infected by *V. vulnificus* for 2 h as described above were obtained. To analyze pro-inflammatory cytokines in serum, murine blood samples collected from *V. vulnificus*-infected 5-week-old female ICR mice at different time points (0, 3, 5, and 7 h post-infection; $n = 20$ per group in total) were kept at room temperature for 4 h. The supernatant was transferred to a new 1.5 mL tube and centrifuged at $1000 \times g$ at room temperature for 20 min to remove red blood cells. Then, cytokines (mouse IL-6, mouse TNF, and mouse MCP-1) in the medium or murine serum were measured using an ELISA kit (BD Biosciences).

## Reactive oxygen species (ROS) detection assay

To detect intracellular ROS, RAW 264.7 cells were treated for 24 h with phorbol 12-myristate 13-acetate (2 µg mL$^{-1}$). Then, the medium was exchanged for DMEM, and the cells were infected with *V. vulnificus* variants as described above (MOI = 5, $5 \times 10^6$ CFU). After incubation for 1.5 h, 20 µM 2′,7′-dichlorofluorescin diacetate (DCFDA; Sigma Aldrich) was added and further incubated for 30 min at 37 °C. The medium was removed and the cells were washed twice using FACS buffer [2% (v/v) FBS and 0.05% NaN$_3$ in PBS]. Then, the cells were resuspended in fixation buffer [0.2% (v/v) formaldehyde in FACS buffer], and

intracellular ROS were analyzed using a BD FACSCalibur® cytometer (BD Biosciences). The obtained data were analyzed by FCS Express™ software (version 7; De Novo Software).

## Immunoblot analysis

BMDMs treated with *V. vulnificus* were lysed in RIPA lysis buffer and centrifuged at $16,000 \times g$ for 30 min at 4 °C. The supernatant was mixed with $5 \times$ SDS-PAGE sample buffer and subjected to denaturing gel electrophoresis (Bolt™ 4–12% Bis-Tris Plus; Invitrogen). After electrophoresis, proteins were transferred to a PVDF membrane, which was then washed three times with TBST buffer [0.1% (v/v) Tween-20 in TBS]. The PVDF membrane was incubated for 1 h at room temperature in blocking buffer [4% (w/v) bovine serum albumin in TBST buffer], and further incubated at 4 °C overnight with blocking buffer containing a primary antibody. The membrane was then washed three times with TBST buffer and incubated again for 1 h at room temperature in blocking buffer containing the secondary antibody. Then, chemiluminescence was visualized using ECL™ Prime western blotting detection reagent (Cytiva) and a Davinch-Chemi™ chemiluminescence imaging system (Davinch-K).

## Antibodies

The following antibodies were used for immunoblotting: anti-NOXA2/p67phox (1:1000; Abcam, ab109366), anti-NOX2/gp91phox (1:1000; Abcam, ab129068), anti-p-MSK2 Thr568 (1:1000; Assay Biotech, A8149), anti-MSK1 (1:1000; R&D Systems, AF2518), anti-p47phox (1:1000; Santa Cruz Biotechnology, sc-17845), anti-MSK2 (1:1000; Santa Cruz Biotechnology, sc-377151), anti-p-NF-kB p65 Ser276 (1:1000; Thermo Fisher Scientific, PA5-37718). The following antibodies were from Cell Signaling Technology: anti-Rac1/Cdc42 (1:1000, #4651), anti-p-ERK1/2 T202/Y204 (1:1000, #9101), anti-ERK1/2 (1:1000, #4695), anti-p-SAPK/JNK T183/Y185 (1:1000, #4668), anti-SAPK/JNK (1:1000, #9252), anti-p-p38 MAPK T180/Y182 (1:1000, #4511), anti-p38 MAPK (1:1000, #8690), anti-p-MSK1 T581 (1:1000, #9595), anti-p-CREB S133 (1:1000, #9198), anti-CREB (1:1000, #9197), anti-p-c-Jun S37 (1:1000, #3270), anti-c-Jun (1:1000, #9165), anti-ac-NF-kB p65 K310 (1:1000, #3045), anti-NF-kB p65 (1:1000, #8242), anti-β-Actin (1:5000, #12620). Details are provided in the Supplementary Data 1.

## Immunoprecipitation assay

To analyze the assembly of the NOX2 complex, BMDMs were infected with the indicated *V. vulnificus* strains for 2 h as described above. Then, the medium was removed and the cells were lysed using RIPA lysis buffer containing a protease and phosphatase inhibitor cocktail (GenDepot). The cell lysates were incubated for 1 h at 4 °C with protein A/G agarose (Santa Cruz Biotechnology) to remove non-specifically bound proteins. Then, supernatants were incubated at 4 °C overnight with an anti-gp91$^{phox}$ antibody, followed by incubation with protein A/G agarose for 4 h at 4 °C. The immunoprecipitates were washed five times in RIPA lysis buffer, and subsequently eluted by adding $5 \times$ SDS-PAGE sample buffer and boiling at 95 °C for 10 min. Finally, the samples were analyzed by SDS-PAGE followed by immunoblotting with anti-Rac1, anti-p47$^{phox}$, anti-p67$^{phox}$, and anti-gp91$^{phox}$ antibodies.

## RNAi silencing of gp91$^{phox}$

RAW 264.7 cells ($1 \times 10^6$) in serum-free media were transfected with siRNA duplex (Supplementary Data 1) using TransIT-TKO transfection reagent (Mirus Bio). After incubation for 6 h, DMEM containing 20% FBS was added at a 1:1 ratio, and further incubated until 24 h for gene silencing. Non-targeting siRNA was used as a control, and the expression level of gp91$^{phox}$ was validated using immunoblot analysis.

## Quantification of bacterial load

To measure the microbial populations in different tissues, liver, spleen, and blood were collected from 5-week-old female ICR mice ($n = 14$ per

group) infected by *V. vulnificus* variants at 10 h post-infection. The weight of collected tissues was measured and lysed using TissueLyser II (30 Hz for 2 min; Qiagen) in 1 mL DPBS. Next, 50 μL of lysed tissue or blood was serially diluted in DPBS, plated on LBS agar supplemented with polymyxin B (100 units mL$^{-1}$), and incubated overnight at 30 °C. Then, the number of bacterial colonies was counted, and normalized to grams (for liver and spleen) or liters (for blood), respectively.

### Histological analysis of murine tissues

Mice were sacrificed at 10 h post-infection with *V. vulnificus*, and liver tissues were harvested and fixed overnight at room temperature with 10% formalin. The fixed tissues were washed in tap water for 8 h and embedded in paraffin blocks on a Tissue Embedding Center module (Sakura Finetek). The paraffin blocks were sectioned to a thickness of 7 μm using a microtome (Sakura Finetek) and deparaffinized in xylene. Liver tissues were further rehydrated in ethanol, and stained with hematoxylin and eosin (H&E). The stained tissue sections were imaged with an optical microscope (Olympus).

### Statistical analysis

Statistical analyses were performed using Prism software (version 6; GraphPad), and *P*-values were calculated using one-way ANOVA with multiple comparisons, the non-parametric Mann-Whitney *U*-test (two-tailed), two-way ANOVA with multiple comparisons, and the log-rank test. All results are presented as the mean ± SEM.

### Reporting summary

Further information on research design is available in the Nature Portfolio Reporting Summary linked to this article.

## Data availability

The atomic models and structure factors have been deposited in the Protein Data Bank under accession codes 8KA2 (RDTND-RID$_{CBD}$), 8KA1 (RDTND-RID$_{CBD}$/CaM), 8K9Z (RDTND-RID$_{CBD}$/CaM/Ca$^{2+}$), and 8KA0 (RDTND$_{E/Q}$-RID$_{CBD}$/CaM/Ca$^{2+}$/NAD$^+$). Crystal structures 1CFD (CaM), 1CLL (CaM/Ca$^{2+}$), NAD$^+$-bound or NAD$^+$-unbound CD38 from human (2I65; 6VUA, 1YH3), 3ZWM (NAD$^+$-bound ADP-ribosyl cyclase from *A. californica*), 1R12 (NAD$^+$-unbound ADP-ribosyl cyclase from *A. californica*), 5XN7 (RID$_{Vv}$), and 3TH5 (active Rac1) referred to in the manuscript can be found on the Protein Data Bank. The cryo-EM density map has been deposited in the Electron Microscopy Data Bank under accession code EMD-37593 (RDTND-RID$_{C/A}$/CaM/Rac1$_{Q/L}$). The mass spectrometry proteomics data generated in this study have been deposited to the ProteomeXchange Consortium via the PRIDE database, with the dataset identifier PXD047927. The amino acid sequences of MARTX$_{Vv}$ RID (NCBI accession ID: WP_015728045.1 [https://www.ncbi.nlm.nih.gov/protein/WP_015728045.1]) and MARTX$_{Vc}$ RID (NCBI accession ID: WP_108347803.1 [https://www.ncbi.nlm.nih.gov/protein/WP_108347803.1]) were downloaded from the NCBI database for the sequence alignment in this study. Source data are provided with this paper.

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

## Acknowledgements

We thank the beamline staff at Pohang Accelerator Laboratory (BL-5C and 11 C) and the cryo-EM facility staff at the Institute of Membrane Proteins (Pohang) and KRIBB, Korea, for their kind help with data collection. Computational analysis of cryo-EM data was performed on the High Performance Computing Resources in the IBS Research Solution Center, Daejeon, Korea. This work was supported by the National Research Foundation of Korea (NRF-2021M3A9I4022934 to M.H.K.) funded by MSIT and by the KRIBB Research Initiative Program (KGM1382413 and KGM9942421 to M.H.K).

## Author contributions

M.H.K. conceived and supervised the work; S.C. designed and carried out crystallization and cryo-EM sample preparation, cell biological, immunological, and animal experiments, with the help of Y.L., S.P., S.-M.K., T.-H.K., and J.H.; Y.L. designed and performed biochemical and structural biological experiments, with the help of S.C., S.Y.J., and D.W.O.; J.P. analyzed amino acid sequences and domains of the MARTX toxins; G.S.L. and J.H.M. performed the LC/MS experiments for interactome analysis; G.S.L., D.L., E.-H.K., H.-K.C., and J.H.M. analyzed by-products of NAD(P)$^+$ using LC/MS and NMR, with the help of S.C. and J.H.; S.C., Y.L., C.C., B.S.K., J.-J.S., J.H., and M.H.K. analyzed and interpreted the data; S.C., Y.L., and J.H. drafted the manuscript, and M.H.K. edited the manuscript, with input from all authors.

## Competing interests

The authors declare no competing interests.

## Additional information

**Peer review information** : *Nature Communications* thanks Takashige Kashimoto and the other, anonymous, reviewers for their contribution to the peer review of this work. A peer review file is available.

