## [Peer Review File · Nature Communications]

Dissemination of pathogenic bacteria is reinforced by a MARTX toxin effector duetREVIEWER COMMENTS

Reviewer #1 (Remarks to the Author):

In the manuscript, Choi et al. report virulent mechanisms for the multifunctional autoprocessing repeats-in-toxin (MARTX) toxin from *Vibrio vulnificus*. After cleavage activated by host co-factor, the resulted domain duet DUF1-RID binds to host calmodulin and the active form of Rac1. Crystal and cryo-EM structures of DUF1-RID bound with calmodulin and/or Rac1 reveal the interaction mechanisms. DUF1 shows structural similarity to NAD⁺ hydrolase and the activity is dependent on the interaction with calmodulin which requires RID. RID functions as an N ϵ -fatty acyltransferase that modifies Rac1. The DUF1-RID duet inhibits host ROS generation by depleting NAD(P)⁺ (DUF1) and inactivating Rac1 (RID). The MARTX toxin paralyzes host immune responses, promotes bacterial dissemination, and induces sepsis. Combining biochemical, biophysical, cell biology, and in vivo studies, the authors illustrated the pathogenic principles for the key toxin from *Vibrio vulnificus*. The research work is rigorous, informative with very convincing replicative data. The manuscript is well written organized and prepared.

Minor comment

An alternative to the statement

“It should be mentioned however that the interaction between the MARTXVv RID and Rac1 (KD = 20 μ M) is significantly weaker than that between the MARTXVc RID and Rac1 (KD = 4 μ M)¹⁴. The weaker interaction between the MARTXVv RID and Rac1 may provide the flexibility necessary to transform RDTND into a NAD(P)⁺-hydrolyzing enzyme via hijacking of CaM. It should be mentioned however that the interaction between the MARTXVv RID and Rac1 (KD = 20 μ M) is significantly weaker than that between the MARTXVc RID and Rac1 (KD = 4 μ M)¹⁴. The weaker interaction between the MARTXVv RID and Rac1 may provide the flexibility necessary to transform RDTND into a NAD(P)⁺-hydrolyzing enzyme via hijacking of CaM.”

..is that affinity may change (be higher) if determined when both membrane bound proteins are in a membrane. Their localization may dictate specificity. Are positive charges in the same place and number for MARTXVv RID and MARTXVc RID? As stated in discussion membrane affinity may play a role.

Reviewer #2 (Remarks to the Author):

Major comment

The authors found that RDTND of *V. vulnificus* MARTX suppresses ROS production by decreasing NAD⁺ levels and reducing acute inflammation, allowing it to disseminate in infected mice.

I think that their results on the protein structure are clear, quite novel, and well-written. However, I think there is a lack of important evidence as to whether or not it was involved in the invasion as written in the title. The most critical concern is whether the MARTX suppresses cytokine production in vivo through the NADase activity of RDTND. The authors showed this only in Fig.6a. The authors should confirm whether the MARTX suppresses cytokine production in vivo through the NADase activity of RDTND using a KO mouse (NOXs KO).

Minor comment

Throughout, the notation of Vc or Vv is very confusing and confusing, and I get the impression that many of the references are based on Vc.

Abstract: "Is it not clear how the toxin induces septicemia" is the subject of this study? I feel that the abstract needs to show more important questions.

Introduction:

The authors should state in the introduction that like Lines 110–111: Among the different genera, *Vibrio* spp. is the primary genus that secretes MARTX toxins.

Title: Although the title says "invasion strategy", many parts are unclear about its role in vivo, so its impression is exaggerated. I recommend using "dissemination".

L53 Upon entry into host cells → It is necessary to clearly indicate whose entry (bacteria or only toxins)

L69 invasion of the host → It is necessary to clearly indicate whose invasion to where (intracellular or other organs?)

L69-70 The references of 12,13 is on *V. cholerae*. The invasion mechanisms of *V. cholera* and *V. vulnificus* are not the same, and the references are talking about intestinal infections. To discuss this, the authors are forced to use an intestinal infection model (but they are using a subcutaneously infection model). They should add more detail about the differences in the mechanisms.

L89 an interactome analyses → interactome analyses

L120 catalytic → a catalytic

L127 Although I can understand that a part of the RID structure is required for the activity of DUF-1 (RDTND) from Fig. 1d, the possibility that DUF-1 alone is active cannot be ruled out. The authors should show that the results with DUF-1 alone as a control in Fig. 1d.

L151 was→were

L195 to further validated → to further validate

L199 fig2G, H, I The authors showed that NAD⁺-degrading activity was completely abolished by every mutation. It's critical, however, it is questionable whether these sites are actually important residues. What should the authors do to clear that? It may be better to show that it is not completely abolished by the mutations into other sites or to create mutants that show intermediate phenotype by substituting amino acids with structures similar to the original residue.

L406-408, L427-429 The reduction of NAD⁺ and causing sepsis in their method (subcutaneous inoculation) depends on the number of bacteria in the circulation, and it has not been proven that it depends on the presence of effectors or mutations.

Authors should evaluate whether MARTX acts directly on PBMC. Is PBMC a target for MARTX? It is necessary to show whether PBMC is a specific target or whether RDTND-RID of MARTX acts on all cells. L442 that unlike NADase-secreting → that, unlike NADase-secreting
L509 Why did the authors do “Vibrio vulnificus MO6-24/O strains were grown at 30°C”? I think 37°C is the optimal degree of V. vulnificus’s growth. They should add the results and references for that.
L660 reservoir → a reservoir
L851 Did the authors use normal PBS? It is known that PBS with normal salt concentration causes V. vulnificus lysis due to osmotic pressure. They should show the NaCl concentration.
L858 Where is the dorsal side? It is a wide range. Authors need to indicate whether the dorsal is center, ventral, cranial, or caudal. It is unclear whether these are appropriate models for this study. The authors need to discuss the proliferation depending on MARTX at the local infected site.
L882 a further → further
L908 DMEM and the cells → DMEM, and the cells
L910 added, → added
L932 assembly → the assembly
L946 were → was
L947 bloods?
Extended data Fig. 8a makes it easier to understand the diagrams of the genetic structure of the mutants. Authors should show it as a main figure.

Reviewer #3 (Remarks to the Author):

MARTX toxin by *Vibrio vulnificus* consists of DUF1 and RID domain.
The function of DUF1 has yet to be discovered. This paper showed that DUF1 acts as a RID-dependent transforming NADase domain (RDTND) that cleaves NAD⁺ to ADP-ribose by recruiting calmodulin. The cryo-EM structure of the RDTND-RID duet complexed with calmodulin and Rac1. Furthermore, the author showed the duet suppresses ROS generation by depleting NAD(P)⁺ and modifying Rac1. The approach to reveal the DUF1 function is exciting and the paper brings novel insight of MARTX toxin. The paper is suitable for the Nature Communications, but I have some questions and comments.

Major points

They found DUF1 has a similar structure to CD38. Did the author check the activity of ADP-ribosyl cyclase? Furthermore, it should be noted that it might have the activity of ADP-ribosylation. Some ADP-ribosyltransferases, such as iota toxin Ia, have NADase activity without substrate protein.

The cryo-EM structure of RDTND-RIDC/A/ CaM/Rac1Q61L is essential, but the resolution is not good (4.3Å). Especially, I feel so in extended data Fig.7.

For the grid preparation, is the sample the crosslinked proteins? I request to improve the resolution using non-crosslinked protein.

Minor points

p13 L298

Entire RID includes RIDmld_c and RIDcd, but not RIDcbd.

Is this right?

p16 l356

What is the PBR?

p16 l358

I am not sure the meaning.

which translocate to the membrane (where NOX2 resides)

Reviewer #1:

In the manuscript, Choi et al. report virulent mechanisms for the multifunctional autoprocessing repeats-in-toxin (MARTX) toxin from *Vibrio vulnificus*. After cleavage activated by host co-factor, the resulted domain duet DUF1-RID binds to host calmodulin and the active form of Rac1. Crystal and cryo-EM structures of DUF1-RID bound with calmodulin and/or Rac1 reveal the interaction mechanisms. DUF1 shows structural similarity to NAD⁺ hydrolase and the activity is dependent on the interaction with calmodulin which requires RID. RID functions as an N ϵ -fatty acyltransferase that modifies Rac1. The DUF1-RID duet inhibits host ROS generation by depleting NAD(P)⁺ (DUF1) and inactivating Rac1 (RID). The MARTX toxin paralyzes host immune responses, promotes bacterial dissemination, and induces sepsis. Combining biochemical, biophysical, cell biology, and in vivo studies, the authors illustrated the pathogenic principles for the key toxin from *Vibrio vulnificus*. The research work is rigorous, informative with very convincing replicative data. The manuscript is well written organized and prepared.

Response: We thank the Reviewer for taking the time to review our manuscript and for providing positive feedback.

Minor comment

An alternative to the statement

“It should be mentioned however that the interaction between the MARTX_{Vv} RID and Rac1 (K_D = 20 μ M) is significantly weaker than that between the MARTX_{Vc} RID and Rac1 (K_D = 4 μ M). The weaker interaction between the MARTX_{Vv} RID and Rac1 may provide the flexibility necessary to transform RDTND into a NAD(P)⁺-hydrolyzing enzyme via hijacking of CaM.”

..is that affinity may change (be higher) if determined when both membrane bound proteins are in a membrane. Their localization may dictate specificity. Are positive charges in the same place and number for MARTX_{Vv} RID and MARTX_{Vc} RID? As stated in discussion membrane affinity may play a role.

Response: We thank the Reviewer for providing this constructive comment. As shown in the figure below (Fig. a), MARTX_{Vv} RID and MARTX_{Vc} RID share a high level of sequence identity (85.6%) and similarity (93.5%). We used AlphaFold3¹ to generate a structural model of the MARTX_{Vc} RID using the template structure of MARTX_{Vv} RID (PDB ID 5XN7). As suggested, we compared positively charged residues on the surfaces of MARTX_{Vv} RID (PDB

ID, 5XN7), MARTX_{VV} RID within the cryo-EM structure (RDTND-RID_{C/A}/CaM/Rac1_{Q61L}), and the modeled MARTX_{VC} RID (Fig. b). Positive residues (marked as closed circles and triangles in MARTX_{VV} RID and MARTX_{VC} RID, respectively in Fig. a) are distributed similarly on the surfaces of the RID proteins (Fig. c). More positive residues occur on the surface of MARTX_{VC} RID than on the surface of MARTX_{VV} RID, particularly in the membrane localization domain (MLD)-containing domain (i.e., MLD_C), suggesting that membrane binding of MARTX_{VC} RID may be stronger than that of MARTX_{VV} RID. The weaker membrane binding of MARTX_{VV} RID may be due the structural flexibility needed to transform RDTND into a NAD(P)⁺-hydrolyzing enzyme via hijacking of CaM, which may be more associated with the weaker interaction between MARTX_{VV} RID and Rac1 ($K_D = 20 \mu\text{M}$) than the interaction between MARTX_{VC} RID and Rac1 ($K_D = 4 \mu\text{M}$). The binding affinities of MARTX_{VC} RID to Rac1 and MARTX_{VV} RID to Rac1 were calculated in solution². Since RID proteins are localized to the membrane and RID-catalyzed Rac1 modification is achieved in the membrane, the binding affinities may be higher between both RID proteins and Rac1 in the membrane, as commented by the Reviewer.

We have added this information to the Results section and supporting data to the revised manuscript (page 15; Supplementary Fig. 10).

a**b****c**
Reviewer #2:

Major comment

The authors found that RDTND of *V. vulnificus* MARTX suppresses ROS production by decreasing NAD⁺ levels and reducing acute inflammation, allowing it to disseminate in infected mice. I think that their results on the protein structure are clear, quite novel, and well-written. However, I think there is a lack of important evidence as to whether or not it was involved in the invasion as written in the title. The most critical concern is whether the MARTX suppresses cytokine production in vivo through the NADase activity of RDTND. The authors showed this only in Fig.6a. The authors should confirm whether the MARTX suppresses cytokine production in vivo through the NADase activity of RDTND using a KO mouse (NOXs KO).

Response: We sincerely thank the Reviewer for giving us the opportunity to think over and address this issue again. Our data clearly showed that both NAD(P)⁺ depletion (Fig. 5b and 5c) and Rac1 modification (Fig. 5f) by the RDTND-RID duet are linked to reduced production of ROS (Fig. 5d and 5e) and suppressed cytokine production (Fig. 5h–j) at the cellular level (i.e., in macrophages). In detail, we engineered *Vibrio vulnificus* to inactivate RID function (i.e., EF::RDTND-RID_{C/A} strain showing only NAD(P)ase function). Using the engineered strain, clearly demonstrated that the NADase activity of RDTND-RID_{C/A} reduces ROS production (Fig. 5b and 5c) and suppresses cytokine production (Fig. 5h–j). Additionally, we showed that RID significantly reduced Rac1 association (critical for NOX2 activation) with key subunits of NOX2 complex, leading to NOX2 inactivation (Fig. 5f). We should mention that we obtained these data from BMDMs that had been infected with *V. vulnificus* strains for 2 h because severe cell rounding and lysis of the cells infected with strains having the NADase function or Rac1 modification function or both functions was observed after that infection time.

The Reviewer suggested that we further validate the suppression of cytokine production *in vivo* by examining only the NADase activity of RDTND using the NOX KO mouse (i.e., excluding RID function). We fully appreciate the concern raised by the Reviewer and seriously considered performing these experiments. However, we experienced several problems in performing these experiments because of the immunological consequences of *V. vulnificus* infection.

As shown in Fig. 6e and 6f, we could not detect cytokines in serum collected from the mice infected with *V. vulnificus* strains including strains having only NADase function (i.e., EF::RDTND-RID_{C/A} strain) or both RNTND and RID functions (i.e., EF::RDTND-RID strain) at

3 h post infection (Fig. 6e and 6f). Also, we observed a cytokine storm, revealing that bacterial sepsis had occurred in mice at 5 h and 7 h post-infection (Fig. 6e and 6f). These data indicate that, because of sepsis, it would be very difficult to decide which time points are suitable for sampling in order to observe and compare cytokines in NOX KO mice (i.e., at the system level).

In addition, many previous reports have demonstrated that NOX defective mice are more susceptible to bacterial infection than wild-type mice. One example is mice with NOX2 inactivated by a spontaneous mutation in p47phox (one of the key components in the NOX2 complex). These mice were more susceptible to systemic challenge with bacteria such as *Staphylococcus xylosus* and *Staphylococcus aureus* than wild-type mice and exhibited higher mortality³. This means that the results of experiments using NOX KO mice will be much more complicated to interpret results because of the problems associated with sepsis. Based on our present data and previously reported results, we carefully reached the conclusion that *in vivo* experiments using NOX KO mice would not be suitable to address the Reviewer's query. Based on these considerations, we sincerely hope that the Reviewer understands the difficulties in performing *in vivo* RDTND function validation using KO mouse.

As an alternative, we generated NOX2-silencing macrophage cells to validate the suppression of cytokine production via only the NADase activity of RDTND. The NOX2-silenced macrophage cells were infected with *V. vulnificus* strains. Overall cytokine levels in the NOX2-silenced macrophage cells infected with the strains were significantly lower than those in wild-type macrophage cells. Nevertheless, the results clearly showed that only NADase activity significantly suppressed pro-inflammatory cytokine production in the NOX2-silenced macrophages infected with the EF::RDTND-RID_{C/A} strain, indicating that depletion of NAD(P)⁺ by RDTND is critically involved in pro-inflammatory cytokine suppression via ROS reduction, independently of RID-mediated inhibition of NOX2 complex formation. We have added these results to the revised manuscript (page 19; Supplementary Fig. 11b-e).

Minor comment

Q1: Throughout, the notation of Vc or Vv is very confusing and confusing, and I get the impression that many of the references are based on Vc.

Response: We apologize for any confusion in the notation of *Vibrio cholerae* (Vc) and *Vibrio vulnificus* (Vv). As described in the manuscript, *Vibrio* spp. including *V. cholerae* and *V. vulnificus* are major species secreting MARTX toxins containing the Rho-inactivation domain (RID) (Fig. 1b). Clinical isolates of *V. cholerae* mostly produce RID-only containing MARTX

toxins while those of *V. vulnificus* generate DUF1-RID containing MARTX toxins (Fig. 1c). As pointed out by the Reviewer, many studies have reported the pathogenicity of the RID derived from *Vc*. The new function of the RDTNT-RID effector duet derived from *Vv* revealed in our study is distinct from that of the *Vc* RID reported in previous studies. Throughout the manuscript, we had to mention *V. cholerae* and *V. vulnificus* many times. This could not be avoided. This is why we needed to use the abbreviations *Vc* and *Vv* for *V. cholerae* and *V. vulnificus*, respectively. Otherwise, the text would have been very laborious to read. We hope that the Reviewer accepts why we had to use the notations *Vc* and *Vv*.

Q2: Abstract: “Is it not clear how the toxin induces septicemia” is the subject of this study? I feel that the abstract needs to show more important questions.

Response: As commented by the Reviewer, we have addressed the issue by adding the text “however, the molecular mechanism via which the toxin contributes to septicemia remains unclear” in the revised manuscript (page 2).

Q3: Introduction:

The authors should state in the introduction that like Lines 110–111: Among the different genera, *Vibrio* spp. is the primary genus that secretes MARTX toxins.

Response: We have added a description to the Introduction in the revised manuscript (page 3).

Q4: Title: Although the title says “invasion strategy”, many parts are unclear about its role in vivo, so its impression is exaggerated. I recommend using “dissemination”.

Response: As suggested, we have changed the title “Invasion strategy of pathogenic bacteria is reinforced by a MARTX toxin effector duet” to “Dissemination of pathogenic bacteria is reinforced by a MARTX toxin effector duet” in the revised manuscript.

Q5: L53 Upon entry into host cells →It is necessary to clearly indicate whose entry (bacteria or only toxins)

Response: As pointed out, we have changed the sentence “Upon entry into host cells,

MARTX toxins are cleaved to release functionally independent effector domains that target specific host substrates” to “Once MARTX toxins translocate into host cells, the toxins are cleaved to release functionally independent effector domains that target specific host substrates” in the revised manuscript (page 3).

Q6: L69 invasion of the host → It is necessary to clearly indicate whose invasion to where (intracellular or other organs?)

Response: Although MARTX toxin-producing pathogens such as *V. vulnificus* are not intracellular pathogens, as described in the manuscript, many studies have reported that secreted MARTX toxins play a crucial role in bacterial evasion or suppression of host immune responses. It has been reported that *V. vulnificus* can exert MARTX toxin-mediated cytotoxicity only after bacteria have made contact with host cells⁴. This study showed that the expression (secretion) of the MARTX toxin increases in a time-dependent manner after the bacteria make contact with the host cell⁴. MARTX toxin-mediated pathogenicity promotes the spread of bacteria to other organs through the bloodstream and subsequently caused septicemia. Based on this background information, we have changed the sentence to “Successful host invasion by pathogenic bacteria such as *V. vulnificus* relies on the ability of their MARTX toxins to counteract host defense mechanisms” in the revised manuscript (page 3).

Q7: L69-70 The references of 12,13 is on *V. cholerae*. The invasion mechanisms of *V. cholerae* and *V. vulnificus* are not the same, and the references are talking about intestinal infections. To discuss this, the authors are forced to use an intestinal infection model (but they are using a subcutaneously infection model). They should add more detail about the differences in the mechanisms.

Response: We thank the Reviewer for this constructive comment. Both *V. cholerae* and *V. vulnificus* cause intestinal infections and are transmitted by consuming contaminated food. *V. vulnificus* can also cause skin infections when open wounds are exposed to contaminated brackish water or uncooked seafood products.

As pointed out the Reviewer, the invasion mechanisms of these two bacteria are not the same, although they share the same effector domains within their MARTX toxins, such as the RID and alpha-beta hydrolase domain (ABH). However, it is clear that *V. cholerae* and *V. vulnificus* share the property of using MARTX toxins to evade the host defense mechanism⁵⁻⁷.

In this study, we emphasized that clinical *V. vulnificus* strains secrete MARTX toxins containing a unique effector duet RDTND-RID that is crucial for suppression of the host immune response. The functional mechanism of the effector duet is distinct from those of typical *V. cholerae* MARTX toxin-driven effectors.

Intraperitoneal infection models have been developed using both *V. cholerae* and *V. vulnificus*, but a wound infection model (i.e., subcutaneous injection, s.c.), which leads to secondary septicemia, has also been developed using *V. vulnificus*, particularly for MARTX toxin studies⁸⁻¹¹.

Q8: L89 an interactome analyses → interactome analyses

Response: We have corrected this in the revised manuscript (page 4).

Q9: L120 catalytic → a catalytic

Response: We have corrected this in the revised manuscript (page 6).

Q10: L127 Although I can understand that a part of the RID structure is required for the activity of DUF-1 (RDTND) from Fig. 1d, the possibility that DUF-1 alone is active cannot be ruled out. The authors should show that the results with DUF-1 alone as a control in Fig. 1d.

Response: As suggested, we performed the experiments with DUF1 again and the updated results are presented in Fig. 1d. The results are referred to in the Results section of the revised manuscript (page 6).

Q11: L151 was→were

Response: We have corrected the typo “was” to “are” in the revised manuscript (page 7).

Q12: L195 to further validated → to further validate

Response: We have corrected the typo “to further validated” to “to further validate” in the revised manuscript (page 9).

Q13: L199 fig2G, H, I The authors showed that NAD⁺-degrading activity was completely abolished by every mutation. It's critical, however, it is questionable whether these sites are actually important residues. What should the authors do to clear that? It may be better to show that it is not completely abolished by the mutations into other sites or to create mutants that show intermediate phenotype by substituting amino acids with structures similar to the original residue.

Response: The results in Fig. 2g–i confirmed that the ability of DUF1-RID_{CBD} to transform DUF1 into a NAD(P)ase depended on its binding to CaM using both DUF1-RID_{CBD} and DUF1-RID_{CBD} complexed with CaM structures. We think that the Reviewer's question relates to the data shown in Fig. 3c. As recommended by the Reviewer, we substituted the residues critical for NAD⁺-hydrolyzing activity, which were selected based on the structure of the NAD⁺-bound RDNTD_{E2186Q}-RID_{CBD}/CaM complex, with structurally similar or smaller residues and re-evaluated the NAD⁺-hydrolyzing activities of the constructs. The results have been updated in the revised manuscript.

Q14: L406-408, L427-429 The reduction of NAD⁺ and causing sepsis in their method (subcutaneous inoculation) depends on the number of bacteria in the circulation, and it has not been proven that it depends on the presence of effectors or mutations.

Response: Fig. 5b and 5c clearly showed that *V. vulnificus* strains secreting MARTX toxin containing RDTND-RID or RDTND-RID_{C/A} decreased NAD(P)⁺ levels in BMDMs, in accordance with the results of biochemical analyses (Fig. 2g–i and 3c), but not the NAD(P)⁺ levels of other mutant strains secreting MARTX toxin containing enzymatically inactive RDTND-RID or RDTND-RID defective for CaM binding. Thus, the NAD(P)⁺ levels in BMDMs infected with *V. vulnificus* strains are the effector duet-dependent consequences. The same was also true for the expression of pro-inflammatory cytokines in BMDMs infected with *V. vulnificus* strains as shown in Fig. 5h–j. Consistent with the results of the biochemical assays (Fig. 2g–i and 3c) and the results in BMDMs cells (Fig. 5b and 5c), NAD⁺ levels were significantly reduced (at 7 h post-infection) in PBMCs of mice infected with *V. vulnificus* strains secreting MARTX toxin containing RDTND-RID or RDTND-RID_{C/A}, but not other mutant strains secreting MARTX toxin containing enzymatically inactive RDTND-RID or RDTND-RID defective for CaM binding (Fig. 6a). Thus, these results further demonstrate that the reduced NAD⁺ levels in mice infected with *V. vulnificus* strains are a consequence of the RDTND-RID effector duet. In accordance with the results, *V. vulnificus* strains secreting MARTX toxin containing RDTND-RID or RDTND-RID_{C/A}, but not mutant strains secreting MARTX toxin

containing enzymatically inactive RDTND-RID or RDTND-RID defective for CaM binding, were disseminated to distal organs of mice at 10 h post-infection (Fig. 6b–d), further supporting the effector duet-dependent events *in vivo*. Collectively, these results clearly indicate that the reduction in the NAD⁺ level *in vivo* leading to sepsis depends on the RDTND-RID effector duet. Many studies have also reported the functions of MARTX toxin effectors *in vivo* by using genome-engineered *Vibrio* strains secreting MARTX toxin containing targeted wild-type and mutant effectors^{2,5,10,12}.

Q15: Authors should evaluate whether MARTX acts directly on PBMC. Is PBMC a target for MARTX? It is necessary to show whether PBMC is a specific target or whether RDTND-RID of MARTX acts on all cells.

Response: PBMCs comprise a variety of immune cells such as lymphocytes, monocytes, and granulocytes, which participate in the immune response to infection¹³⁻¹⁶. When infected with bacteria, the host's immune cells are recruited to the infection site through the bloodstream and kill the bacteria¹⁷. Thus, we evaluated NAD⁺ levels in PBMCs isolated from blood. PBMCs are not the only target of MARTX toxins since MARTX toxins enter host cells regardless of cell type including immune cells and epithelial cells¹⁸.

Q16: L442 that unlike NADase-secreting → that, unlike NADase-secreting

Response: We have corrected “that unlike NADase-secreting” to “that, unlike NADase-secreting” in the revised manuscript (page 22).

Q17: L509 Why did the authors do “*Vibrio vulnificus* MO6-24/O strains were grown at 30°C”? I think 37°C is the optimal degree of *V. vulnificus*'s growth. They should add the results and references for that.

Response: *V. vulnificus* strains are found in warmer estuarine and marine environments where temperatures range from 9 to 31°C¹⁹. Thus, *V. vulnificus* is grown at 30°C, as reported in many epidemiology studies of *V. vulnificus* infections^{10,20-22}. We have added related references in the revised manuscript (page 25).

Q18: L660 reservoir → a reservoir

Response: We have corrected “reservoir” to “a reservoir” in the revised manuscript (page 31).

Q19: L851 Did the authors use normal PBS? It is known that PBS with normal salt concentration causes *V. vulnificus* lysis due to osmotic pressure. They should show the NaCl concentration.

Response: PBS contains 137 mM sodium chloride, 2.7 mM potassium chloride, 10 mM sodium phosphate, and 1.8 mM potassium phosphate. *V. vulnificus* did not lyse by osmotic pressure in PBS, and we also plated serially diluted bacterial suspensions on LBS agar after *in vitro* or *in vivo* experiments to count the number of bacteria. Many studies have used PBS to dilute bacterial suspensions or as a control treatment for *V. vulnificus* infection^{10,22-25}.

Q20: L858 Where is the dorsal side? It is a wide range. Authors need to indicate whether the dorsal is center, ventral, cranial, or caudal. It is unclear whether these are appropriate models for this study. The authors need to discuss the proliferation depending on MARTX at the local infected site.

Response: We apologize for this confusion. The bacterial suspension was subcutaneously injected into the region beneath the skin over the back of the neck. This site is commonly used for subcutaneous injection in *V. vulnificus* infection experiments^{10,12,26-28}. As aforementioned, *V. vulnificus* causes intestinal infections and is transmitted by consuming contaminated food. *V. vulnificus* can also cause skin infections when open wounds are exposed to contaminated brackish water or uncooked seafood products.

The wound infection model (i.e., s.c.), which leads to secondary septicemia, has also been used to study *V. vulnificus* infections, and particularly MARTX toxins⁸⁻¹¹. We have clarified the s.c. injection site in the revised manuscript (page 41).

A previous study using an s.c. infection mouse model showed that *V. vulnificus* MARTX is a critical virulence factor required for bacterial survival and colonization of mice upon infection²⁹. The bacterial population increased time-dependently at the subcutaneously infected site on the back of mouse. The tissue surrounding the infected area revealed disintegrated myofibrils and necrotic adipocytes. Proliferation and dissemination of the bacteria to internal organs through the bloodstream was observed at 9 h post-infection²⁹. *V. vulnificus* lacking the MARTX toxin was defective in conferring cytotoxicity to immune cells and was sensitive to phagocytosis by immune cells, as evidenced by the subsequent reduction in colonization and dissemination of the bacteria to internal organs²⁹. Consistent with a previous study²⁹, our study

also found that pro-inflammatory cytokine levels were up-regulated time-dependently (Fig. 6e, f), and that bacterial loads appeared in the blood, spleen, and liver at 10 h post-infection (Fig. 6b–d), suggesting that the subcutaneously infected *V. vulnificus* RDTND-RID strain or RDTND-RID_{C/A} strain may colonize and subsequently spread to distal organs through the bloodstream. Consequently, the results of the *in vitro* and *in vivo* analyses in our study demonstrated that the RDTND-RID effector duet within the MARTX toxin facilitates the survival and proliferation of *V. vulnificus* at the infection site and its systemic dissemination to internal organs in mice via both its NAD(P)ase and N^ε-fatty acylation enzymatic functions. As recommend by the Reviewer, we have discussed these points in the Results section (RDTND-RID promotes bacterial dissemination and sepsis in mice) of the revised manuscript (page 21).

Q21: L882 a further → further

Response: We have corrected this typo in the revised manuscript (page 43).

Q22: L908 DMEM and the cells → DMEM, and the cells

Response: We have corrected this in the revised manuscript (page 44).

Q23: L910 added, → added

Response: We have corrected this in the revised manuscript (page 44).

Q24: L932 assembly → the assembly

Response: We have corrected this in the revised manuscript (page 45).

Q25: L946 were→was

Response: We have corrected this typo in the revised manuscript (page 46).

Q26: L947 bloods?

Response: We have corrected “bloods” to “blood” in the revised manuscript (page 46).

Q27: Extended data Fig. 8a makes it easier to understand the diagrams of the genetic structure of the mutants. Authors should show it as a main figure.

Response: As suggested, we have moved the diagrams of the genetic structure of the mutants in Extended data Fig. 8a to main Fig. 5a in the revised manuscript.

Reviewer #3:

MARTX toxin by *Vibrio vulnificus* consists of DUF1 and RID domain. The function of DUF1 has yet to be discovered. This paper showed that DUF1 acts as a RID-dependent transforming NADase domain (RDTND) that cleaves NAD⁺ to ADP-ribose by recruiting calmodulin. The cryo-EM structure of the RDTND-RID duet complexed with calmodulin and Rac1. Furthermore, the author showed the duet suppresses ROS generation by depleting NAD(P)⁺ and modifying Rac1. The approach to reveal the DUF1 function is exciting and the paper brings novel insight of MARTX toxin. The paper is suitable for the Nature Communications, but I have some questions and comments.

Response: We deeply thank the Reviewer for taking the time to review our manuscript and for providing positive feedback.

Major points

Q1: They found DUF1 has a similar structure to CD38. Did the author check the activity of ADP-ribosyl cyclase? Furthermore, it should be noted that it might have the activity of ADP-ribosylation. Some ADP-ribosyltransferases, such as iota toxin Ia, have NADase activity without substrate protein.

Response: We appreciate the Reviewer for providing these constructive comments. As suggested, we checked the ADP-ribosyl cyclase activity of RDTND complexed with RID/CaM using a modified HPLC-UV method³⁰. Consistent with previously reported results³¹⁻³³, the analyses showed that CD38 mostly hydrolyzes NAD⁺ to ADPR (99.28%) and generates a small amount of cADPR (0.74%) (Supplementary Fig. 5). RDTND complexed with RID/CaM hydrolyzed NAD⁺ to ADPR (99.69%) and cADPR (0.31%). As described in the manuscript, unlike CD38 and *Aplysia californica* ADP-ribosyl cyclase, RDTND complexed with RID/CaM has a smaller and narrower pocket due to the bulky residues R2234 and Y2181, which obstruct

rotation of the adenine ring during the ADPR cyclization reaction (Fig. 3d). The structural restriction on the movement of NAD⁺ moieties may lead to the RDTND-RID/CaM complex having lower ADP-ribosyl cyclase activity than that of CD38. Frankly speaking, we are not sure whether RDTND complexed with RID/CaM functions as a ADP-ribosyl cyclase, considering the very low amount of cADPR (0.31%) produced relative to the large amount of ADPR produced (99.69%). We have added these results (Supplementary Fig. 5) and related discussions to the revised manuscript (page 12).

ADP-ribosyl transferases (such as iota-toxin) having NADase activity commonly share conserved structural motifs composed of R-S-E or H-Y-E residues in the active site³⁴⁻³⁶. We did not find any similar motifs in the NAD⁺-bound RDNTD_{E2186Q}-RID_{CBD}/CaM complex structure. Additionally, an analysis result of the structural similarity with RDNTD within the complex did not include any ADP-ribosyl transferase structures, suggesting that there is a large difference in the overall structures between ADP-ribosyl transferases and the RDNTD/CaM complex.

Q2: The cryo-EM structure of RDTND-RIDC/A/ CaM/Rac1Q61L is essential, but the resolution is not good (4.3Å). Especially, I feel so in extended data Fig.7. For the grid preparation, is the sample the crosslinked proteins? I request to improve the resolution using non-crosslinked protein.

Response: We initially tried to determine the cryo-EM structure of the non-crosslinked native complex (RDTND-RID_{C/A}/CaM/Rac1_{Q61L}) using various vitrification conditions, but single particle cryo-EM images typically resulted in low-resolution 2D class averages indicating flexible movement (Supplementary Fig. 8), which was probably due to the low binding affinity between RID and Rac1. Thus, we applied the cross-linking method to reduce flexible movement in the sample and the overall resolution of the cryo-EM map improved to ~4.3 Å. To facilitate understanding, we have described the cryo-EM experiments and the results obtained using the native complex sample in the revised manuscript (page 13).

Minor points

Q1: p13 L298, Entire RID includes RIDmld_c and RIDcd, but not RIDcbd. Is this right?

Response: Based on the results of sequence analysis of DUF1-RID (i.e., RDTND-RID) within MARTX toxins along with cysteine protease domain (CPD)-mediated cleavage analysis³⁷ and structural analysis of MARTX_{VV} RID², we confirmed that the RID is composed of an N-terminal

domain (RID_{ND}), a membrane localization domain (MLD)-containing domain (RID_{MLD_C}), and a catalytic domain RID (RID_{CD}). We confirmed that nearly all RIDs from *Vibrio* spp. have the same domain structures, with few exceptions (Fig. 1a and 1b). We found that the RID_{ND} is the domain responsible for the interaction of CaM with RDTND and we named the RID_{ND} the “CaM-binding domain (RID_{CBD})”. Thus, the entire RID consists of RID_{CBD}, RID_{MLD_C}, and RID_{CD}.

Q2: p16 l356, What is the PBR?

Response: PBR is the C-terminal polybasic region of Rac1. We have abbreviated it to “PBR” as described in the manuscript.

Q3: p16 l358, I am not sure the meaning. which translocate to the membrane (where NOX2 resides)

Response: We have clarified the sentence “NOX2 activation is mediated by four cytosolic subunits (p47^{phox}, p67^{phox}, Rac1 or Rac2, and p40^{phox}), which translocate to the membrane (where NOX2 resides) to form the active NOX2 complex” to “The transmembrane gp91^{phox} catalytic subunit (i.e., NOX2) is activated by the assembly of four cytosolic subunits (p47^{phox}, p67^{phox}, Rac1 or Rac2, and p40^{phox}), which translocate to the membrane to form the active NOX2 complex” in the revised manuscript (page 17).

- 1 Abramson, J. *et al.* Accurate structure prediction of biomolecular interactions with AlphaFold 3. *Nature* (2024). <https://doi.org:10.1038/s41586-024-07487-w>
- 2 Zhou, Y. *et al.* N(epsilon)-Fatty acylation of Rho GTPases by a MARTX toxin effector. *Science* **358**, 528-531 (2017). <https://doi.org:10.1126/science.aam8659>
- 3 Pizzolla, A. *et al.* Reactive oxygen species produced by the NADPH oxidase 2 complex in monocytes protect mice from bacterial infections. *J Immunol* **188**, 5003-5011 (2012). <https://doi.org:10.4049/jimmunol.1103430>
- 4 Kim, Y. R. *et al.* *Vibrio vulnificus* RTX toxin kills host cells only after contact of the bacteria with host cells. *Cell Microbiol* **10**, 848-862 (2008). <https://doi.org:10.1111/j.1462-5822.2007.01088.x>
- 5 Woida, P. J. & Satchell, K. J. F. The *Vibrio cholerae* MARTX toxin silences the inflammatory response to cytoskeletal damage before inducing actin cytoskeleton collapse. *Sci Signal* **13** (2020). <https://doi.org:10.1126/scisignal.aaw9447>
- 6 Jeong, H. G. & Satchell, K. J. Additive function of *Vibrio vulnificus* MARTX(Vv) and VvHA cytolytins promotes rapid growth and epithelial tissue necrosis during intestinal infection.

- PLoS Pathog* **8**, e1002581 (2012). <https://doi.org:10.1371/journal.ppat.1002581>
- 7 Kim, B. S. *et al.* MARTX Toxin-Stimulated Interplay between Human Cells and *Vibrio vulnificus*. *mSphere* **5** (2020). <https://doi.org:10.1128/mSphere.00659-20>
- 8 Yamazaki, K., Kashimoto, T., Kado, T., Yoshioka, K. & Ueno, S. Increased Vascular Permeability Due to Spread and Invasion of *Vibrio vulnificus* in the Wound Infection Exacerbates Potentially Fatal Necrotizing Disease. *Front Microbiol* **13**, 849600 (2022). <https://doi.org:10.3389/fmicb.2022.849600>
- 9 Ziolo, K. J. *et al.* *Vibrio vulnificus* biotype 3 multifunctional autoprocessing RTX toxin is an adenylate cyclase toxin essential for virulence in mice. *Infect Immun* **82**, 2148-2157 (2014). <https://doi.org:10.1128/IAI.00017-14>
- 10 Lee, Y. *et al.* Makes caterpillars floppy-like effector-containing MARTX toxins require host ADP-ribosylation factor (ARF) proteins for systemic pathogenicity. *Proc Natl Acad Sci U S A* **116**, 18031-18040 (2019). <https://doi.org:10.1073/pnas.1905095116>
- 11 Oliver, J. D. Wound infections caused by *Vibrio vulnificus* and other marine bacteria. *Epidemiol Infect* **133**, 383-391 (2005). <https://doi.org:10.1017/s0950268805003894>
- 12 Choi, S., Kim, B. S., Hwang, J. & Kim, M. H. Reduced virulence of the MARTX toxin increases the persistence of outbreak-associated *Vibrio vulnificus* in host reservoirs. *J Biol Chem* **296**, 100777 (2021). <https://doi.org:10.1016/j.jbc.2021.100777>
- 13 Autissier, P., Soulas, C., Burdo, T. H. & Williams, K. C. Evaluation of a 12-color flow cytometry panel to study lymphocyte, monocyte, and dendritic cell subsets in humans. *Cytometry Part A* **77A**, 410-419 (2010). <https://doi.org:https://doi.org/10.1002/cyto.a.20859>
- 14 Haller, D., Blum, S., Bode, C., Hammes, W. P. & Schiffrin, E. J. Activation of Human Peripheral Blood Mononuclear Cells by Nonpathogenic Bacteria In Vitro: Evidence of NK Cells as Primary Targets. *Infection and Immunity* **68**, 752-759 (2000). <https://doi.org:doi:10.1128/iai.68.2.752-759.2000>
- 15 José Luis Muñoz, C., Flor Pamela Castro, G., Oscar Gutiérrez, C., María Alejandra Moreno, G. & Juan Francisco Contreras, C. in *Physiology and Pathology of Immunology* (ed Rezaei Nima) Ch. 6 (IntechOpen, 2017).
- 16 Bossel Ben-Moshe, N. *et al.* Predicting bacterial infection outcomes using single cell RNA-sequencing analysis of human immune cells. *Nature Communications* **10**, 3266 (2019). <https://doi.org:10.1038/s41467-019-11257-y>
- 17 Shi, C. & Pamer, E. G. Monocyte recruitment during infection and inflammation. *Nature Reviews Immunology* **11**, 762-774 (2011). <https://doi.org:10.1038/nri3070>
- 18 Satchell, K. J. Multifunctional-autoprocessing repeats-in-toxin (MARTX) Toxins of Vibrios. *Microbiol Spectr* **3** (2015). <https://doi.org:10.1128/microbiolspec.VE-0002-2014>
- 19 Strom, M. S. & Paranjpye, R. N. Epidemiology and pathogenesis of *Vibrio vulnificus*. *Microbes Infect* **2**, 177-188 (2000). [https://doi.org:10.1016/s1286-4579\(00\)00270-7](https://doi.org:10.1016/s1286-4579(00)00270-7)
- 20 Woida, P. J., Kitts, G., Shee, S., Godzik, A. & Satchell, K. J. F. Actin Cross-Linking Effector Domain of the *Vibrio vulnificus* F-Type MARTX Toxin Dominates Disease Progression During

- Intestinal Infection. *Infect Immun* **90**, e0062721 (2022). <https://doi.org:10.1128/iai.00627-21>
- 21 Chung, H. Y., Bian, Y., Lim, K. M., Kim, B. S. & Choi, S. H. MARTX toxin of *Vibrio vulnificus* induces RBC phosphatidylserine exposure that can contribute to thrombosis. *Nat Commun* **13**, 4846 (2022). <https://doi.org:10.1038/s41467-022-32599-0>
- 22 Hwang, J. *et al.* Structural insights into the regulation of sialic acid catabolism by the *Vibrio vulnificus* transcriptional repressor NanR. *Proc Natl Acad Sci U S A* **110**, E2829-2837 (2013). <https://doi.org:10.1073/pnas.1302859110>
- 23 Murciano, C. *et al.* MARTX Toxin in the Zoonotic Serovar of *Vibrio vulnificus* Triggers an Early Cytokine Storm in Mice. *Front Cell Infect Microbiol* **7**, 332 (2017). <https://doi.org:10.3389/fcimb.2017.00332>
- 24 Chen, C. L. *et al.* *Vibrio vulnificus* MARTX cytotoxin causes inactivation of phagocytosis-related signaling molecules in macrophages. *J Biomed Sci* **24**, 58 (2017). <https://doi.org:10.1186/s12929-017-0368-2>
- 25 Cho, C., Choi, S., Kim, M. H. & Kim, B. S. *Vibrio vulnificus* PlpA facilitates necrotic host cell death induced by the pore forming MARTX toxin. *J Microbiol* **60**, 224-233 (2022). <https://doi.org:10.1007/s12275-022-1448-x>
- 26 Kim, I. H. *et al.* *Vibrio vulnificus* Secretes an Insulin-degrading Enzyme That Promotes Bacterial Proliferation in Vivo. *J Biol Chem* **290**, 18708-18720 (2015). <https://doi.org:10.1074/jbc.M115.656306>
- 27 Choi, Y., Shin, E., Lee, M., Yeom, J. H. & Lee, K. Functional conservation of specialized ribosomes bearing genome-encoded variant rRNAs in *Vibrio* species. *PLoS One* **18**, e0289072 (2023). <https://doi.org:10.1371/journal.pone.0289072>
- 28 Lee, Z. W. *et al.* Small-molecule inhibitor of HlyU attenuates virulence of *Vibrio* species. *Sci Rep* **9**, 4346 (2019). <https://doi.org:10.1038/s41598-019-39554-y>
- 29 Lo, H.-R. *et al.* RTX Toxin Enhances the Survival of *Vibrio vulnificus* During Infection by Protecting the Organism From Phagocytosis. *The Journal of Infectious Diseases* **203**, 1866-1874 (2011). <https://doi.org:10.1093/infdis/jir070>
- 30 Balducci, E. *et al.* Assay methods for nicotinamide mononucleotide adenyltransferase of wide applicability. *Anal Biochem* **228**, 64-68 (1995). <https://doi.org:10.1006/abio.1995.1315>
- 31 Howard, M. *et al.* Formation and hydrolysis of cyclic ADP-ribose catalyzed by lymphocyte antigen CD38. *Science* **262**, 1056-1059 (1993). <https://doi.org:10.1126/science.8235624>
- 32 Takasawa, S. *et al.* Synthesis and hydrolysis of cyclic ADP-ribose by human leukocyte antigen CD38 and inhibition of the hydrolysis by ATP. *J Biol Chem* **268**, 26052-26054 (1993).
- 33 Shrimp, J. H. *et al.* Revealing CD38 cellular localization using a cell permeable, mechanism-based fluorescent small-molecule probe. *J Am Chem Soc* **136**, 5656-5663 (2014). <https://doi.org:10.1021/ja411046j>
- 34 Sakurai, J., Nagahama, M., Oda, M., Tsuge, H. & Kobayashi, K. Clostridium perfringens iota-toxin: structure and function. *Toxins (Basel)* **1**, 208-228 (2009). <https://doi.org:10.3390/toxins1020208>

- 35 Yoshida, T. & Tsuge, H. Common Mechanism for Target Specificity of Protein- and DNA-Targeting ADP-Ribosyltransferases. *Toxins (Basel)* **13** (2021). <https://doi.org:10.3390/toxins13010040>
- 36 Roussin, M. & Salcedo, S. P. NAD⁺-targeting by bacteria: an emerging weapon in pathogenesis. *FEMS Microbiol Rev* **45** (2021). <https://doi.org:10.1093/femsre/fuab037>
- 37 Shen, A. *et al.* Mechanistic and structural insights into the proteolytic activation of *Vibrio cholerae* MARTX toxin. *Nat Chem Biol* **5**, 469-478 (2009). <https://doi.org:10.1038/nchembio.178>

REVIEWERS' COMMENTS

Reviewer #1 (Remarks to the Author):

Satisfied reviewers concerns.

Reviewer #2 (Remarks to the Author):

We appreciate your explanation regarding the challenges of in vivo experiments using NOx knockout (KO) mice. The detailed discussion on the susceptibility of NOx-deficient mice to bacterial infections and complications due to sepsis highlights the difficulties in interpreting such experimental results.

While it has been pointed out that in vivo verification of RDTND function using KO mice is not feasible, we understand and accept the reasons behind this decision. The alternative approach using NOX2-silenced macrophage cells effectively addresses this question and provides substantial evidence that the NADase activity of RDTND plays a critical role in suppressing inflammatory cytokine production independently of NOX2 inhibition via RID.

The additional results and data included in the revised manuscript adequately address our concerns and contribute to the robustness of the findings.

Reviewer #3 (Remarks to the Author):

MARTX toxin by *Vibrio vulnificus* consists of DUF1 and RID domain. The function of DUF1 has yet to be discovered. This paper showed that DUF1 acts as a RID-dependent transforming NADase domain (RDTND) that cleaves NAD⁺ to ADP-ribose by recruiting calmodulin. The cryo-EM structure of the RDTND-RID duet complexed with calmodulin and Rac1. Furthermore, the author showed the duet suppresses ROS generation by depleting NAD(P)⁺ and modifying Rac1. The approach to reveal the DUF1 function is exciting and the paper brings novel insight of MARTX toxin.

I had some questions and comments.

However, in new version, the manuscript was revised properly.

I think that the paper is suitable for the Nature Communications.